# Stream-level Flow Matching with Gaussian Processes

**Ganchao Wei** [1]   **Li Ma** [1]

## Abstract

Flow matching (FM) is a family of training algorithms for fitting continuous normalizing flows (CNFs). Conditional flow matching (CFM) exploits the fact that the marginal vector field of a CNF can be learned by fitting least-squares regression to the conditional vector field specified given one or both ends of the flow path. In this paper, we extend the CFM algorithm by defining conditional probability paths along "streams", instances of latent stochastic paths that connect data pairs of source and target, which are modeled with Gaussian process (GP) distributions. The unique distributional properties of GPs help preserve the "simulation-free" nature of CFM training. We show that this generalization of the CFM can effectively reduce the variance in the estimated marginal vector field at a moderate computational cost, thereby improving the quality of the generated samples under common metrics. Additionally, adopting the GP on the streams allows for flexibly linking multiple correlated training data points (e.g., time series). We empirically validate our claim through both simulations and applications to image and neural time series data.

## 1. Introduction

Deep generative models aim to estimate and sample from an unknown probability distribution. Continuous normalizing flows (CNFs, Chen et al. (2018)) construct an invertible and differentiable mapping, using neural ordinary differential equations (ODEs), between a source and the target distribution. However, traditionally, it has been difficult to scale CNF training to large datasets (Chen et al., 2018; Grathwohl et al., 2019; Onken et al., 2021). Recently, Lipman et al. (2023); Albergo & Vanden-Eijnden (2023); Liu et al. (2023) showed that CNFs can be trained via a regression

objective and proposed the flow matching (FM) algorithm. FM exploits the fact that the marginal vector field inducing a desired CNF can be learned through a regression formulation, approximating per-sample conditional vector fields using a smoother such as a deep neural network (Lipman et al., 2023). In the original FM approach, the training objective is conditioned on samples from the target distribution, and the source distribution has to be Gaussian. This limitation was later relaxed, allowing the target distribution to be supported on manifolds (Chen & Lipman, 2024) and the source distribution to be non-Gaussian (Pooladian et al., 2023). Tong et al. (2024) provided a unifying framework with arbitrary transport maps by conditioning on both ends. While their framework is general, its application requires the induced conditional probability paths to be readily sampled from, and as such, they considered several Gaussian probability paths. Moreover, most existing FM methods only consider the inclusion of two endpoints, and hence cannot accommodate data involving multiple correlated observations, such as time series and data with a grouping structure. Notably, Albergo et al. (2024) recently proposed multimarginal stochastic interpolants, which aim to learn a multivariate distribution based on correlated observations.

In this paper, we go one level deeper in Bayesian hierarchical modeling of FM algorithm and specify distributional assumptions on *streams*, which are latent stochastic paths connecting the two endpoints. Our approach extends the stochastic interpolant framework proposed by Albergo & Vanden-Eijnden (2023); Albergo et al. (2023), and it leads to a class of CFM algorithms that condition at the "stream" level, which broadens the range of conditional probability paths allowed in CFM training. By endowing the streams with Gaussian process (GP) distributions, these GP-CFM algorithms provide a smoother marginal vector field and wider sampling coverage over its support. Furthermore, conditioning on GP streams allows for flexible integration of correlated observations through placing them along the streams between two endpoints and for incorporating additional prior information, while maintaining the analytical tractability and computational efficiency of CFM algorithms.

In summary, the main contributions of this paper are as follows.

1. We generalize CFM training by augmenting the spec-

---

[1]Department of Statistical Science, Duke University, Durham, NC 27708, USA. Correspondence to: Ganchao Wei <ganchao.wei@duke.edu>, Li Ma <li.ma@duke.edu>.

*Proceedings of the 42nd International Conference on Machine Learning*, Vancouver, Canada. PMLR 267, 2025. Copyright 2025 by the author(s).

ification of conditional probability paths through latent variable modeling on the streams. We show that streams endowed with GP distributions lead to a simple stream-level CFM algorithm that preserves the "simulation-free" training.

2. We demonstrate that appropriately specified GP streams can lead to smoother marginal vector fields and reduced variance in marginal vector estimation, and thereby generate higher quality samples.

3. We show that the GP-based stream-level FM can readily accommodate correlated observations. This allows FM training to borrow information across training samples, thereby improving the marginal vector field estimation and enhancing the quality of the generated samples. Our approach offers additional flexibility to accommodate designs that are challenging for existing approaches, such as allowing correlated observations to be observed along irregularly spaced time points.

4. These benefits are illustrated by simulations and applications to image (CIFAR-10, MNIST and HWD+) and neural time series data (LFP), with code for Python implementation available at https://github.com/weigcdsb/GP-CFM.

## 2. Background and Notation

We start by reviewing the necessary background and defining the notation for flow matching (FM). At the end of this section, we briefly present a Bayesian decision-theoretic perspective on FM training, providing an additional justification for FM algorithms beyond gradient matching (more details in Appendix A).

Consider i.i.d. training observations from an unknown population distribution $q_1$ over $\mathbb{R}^d$. A CNF is a time-dependent diffeomorphic map $\phi_t$ that transforms a random variable $x_0 \in \mathbb{R}^d$ from a source distribution $q_0$ into a random variable from $q_1$. The CNF induces a distribution of $x_t = \phi_t(x_0)$ at each time $t$, which is denoted by $p_t$, thereby forming a probability path $\{p_t : 0 \leq t \leq 1\}$. This probability path should (at least approximately) satisfy the boundary conditions $p_0 = q_0$ and $p_1 = q_1$. It is related to the flow map through the change-of-variable formula or the push-forward equation

$$p_t = [\phi_t]_* p_0.$$

FM aims at learning the corresponding vector field $u_t(x)$, which induces the probability path over time by satisfying the continuity equation (Villani, 2008).

The key observation underlying FM algorithms is that the vector field $u_t(x)$ can be written as a conditional expectation involving a conditional vector field $u_t(x|z)$, which induces a conditional probability path $p_t(\cdot|z)$ corresponding to the

conditional distribution of $\phi_t(x)$ given $z$. Here, $z$ is the conditioning latent variable, which can be the target sample $x_1$ (e.g., Ho et al. (2020); Song et al. (2021b); Lipman et al. (2023),) or a pair of $(x_0, x_1)$ on source and target distribution (e.g., Liu et al. (2023); Tong et al. (2024)). Specifically, Tong et al. (2024), generalizing the result from Lipman et al. (2023), showed that

$$u_t(x) = \int u_t(x|z)\frac{p_t(x|z)q(z)}{p_t(x)}dz = \mathbb{E}\left(u_t(x|z)|x_t = x\right),$$

where the expectation is taken over $z$, which one can recognize is the conditional expectation of $u_t(x|z)$ conditional on the event that $x_t = x$. The integral is with respect to the conditional distribution of $z$ given $x_t = x$.

The FM algorithm is motivated by the goal of approximating the marginal vector field $u_t(x)$ through a smoother $v_t^\theta$ (typically a neural network), via the objective

$$\mathcal{L}_{\text{FM}}(\theta) = \mathbb{E}_{t\sim U(0,1),x\sim p_t(x)}\|v_t^\theta(x) - u_t(x)\|^2.$$

In the following, we follow earlier works and assume $t \sim U(0, 1)$ though the algorithms discussed are valid for other sampling distributions of $t$ as well. The FM objective is not identifiable due to the non-uniqueness of the marginal vector fields that satisfy the boundary conditions without further constraints. FM algorithms address this by fitting $v_t^\theta$ to the conditional vector field $u_t(x|z)$ after further specifying the distribution of $q(z)$ along with the conditional probability path $p_t(x|z)$, through minimizing the finite-sample version of the marginal squared error loss. The corresponding loss is referred to as the conditional flow matching (CFM) objective

$$\mathcal{L}_{\text{CFM}}(\theta) = \mathbb{E}_{t\sim U(0,1),z\sim q(z),x\sim p_t(x|z)}\|v_t^\theta(x) - u_t(x|z)\|^2.$$

Traditionally, optimizing the CFM objective is justified because it has the same gradients w.r.t. $\theta$ to the corresponding FM loss (Lipman et al., 2023; Tong et al., 2024). In Appendix A we detail another justification for CFM without involving the gradient-matching argument. In particular, we view this algorithm from a Bayesian estimation perspective and show that approximating the conditional vector field by minimizing the marginal squared error loss ($\mathcal{L}_{\text{FM}}$) can be interpreted as computing the "posterior expectation" of $u_t(x|z)$ under a prior-likelihood setup. This is the Bayes rule under square error loss and is exactly equal to $u_t(x)$.

This Bayesian estimation justification holds for any coherently specified probability model $q(z)$. So long as the conditional probability path $p_t(x|z)$ is tractable, a suitable CFM algorithm can be designed. Therefore, one can enrich the specification of $q(z)$ using Bayesian latent variable modeling strategies. This motivates us to generalize CFM training to the stream level, which we describe in the next section.

# 3. Stream-level Flow Matching

## 3.1. A Per-stream Perspective on Flow Matching

A stream $s$ is a stochastic process $s = \{s_t : 0 \leq t \leq 1\}$, where each $s_t$ is a random variable in the sample space of the training data. We focus on streams connecting one end $x_0$ in the source to the other $x_1$ in the target. From here on, $s$ will take the place of the latent quantity $z$.

Instead of defining a conditional probability path and vector field given one endpoint at $t = 1$ (Lipman et al., 2023) or two endpoints at $t = 0$ and $1$ (Tong et al., 2024), we shall consider defining it given the whole stream connecting the two ends. To achieve this, we need to specify a probability model for $s$. This can be separated into two parts—the marginal model on the endpoints $\pi(x_0, x_1)$ and the conditional model for $s$ given the two ends. That is

$$(x_0, x_1) \sim \pi \quad \text{and} \quad s|s_0 = x_0, s_1 = x_1 \sim p_s(\cdot|x_0, x_1). \tag{1}$$

Our model and algorithm will generally apply to any choice of coupling distribution $\pi$ that satisfies the boundary condition, including, for example, the popular OT-CFM and Schrödinger bridge (entropy-regularized)-CFM, considered in Tong et al. (2024). We defer the description of the specific choices of $p_s(\cdot|x_0, x_1)$ to the next section and for now focus on describing the general framework.

Given a stream $s$, the "per-stream" vector field $u_t(x|s)$ represents the "velocity" (or derivative) of the stream at time $t$, conditional on the event that $s_t = x$, i.e., the stream $s$ passes through $x$ at time $t$. Assuming that the stream is differentiable within time, the per-stream vector field is

$$u_t(x|s) := \dot{s}_t = ds_t/dt,$$

which is defined only on all pairs of $(t, x)$ that satisfy $s_t = x$. See Appendix B for a more detailed discussion on how the per-stream perspective relates to the per-sample perspective on FM.

While the endpoint of the stream $s_1 = x_1$ is an actual observation in the training data, for the task of learning the marginal vector field $u_t(x)$, one can think of our "data" as the event that a stream $s$ passes through a point $x$ at time $t$, that is $s_t = x$. Under the squared error loss, the Bayes estimate for the per-stream conditional vector field $u_t(x|s)$ will be the "posterior" expectation given the "data", which is exactly the marginal vector field

$$u_t(x) = \mathbb{E}(u_t(x|s)|s_t = x) = \mathbb{E}(\dot{s}_t|s_t = x). \tag{2}$$

Following Theorem 3.1 in Tong et al. (2024), we can show that the marginal vector $u_t(x)$ indeed generates the probability path $p_t(x)$. (See the proof in the Appendix J.1.) The essence of the proof is to check the continuity equation for the (degenerate) conditional probability path $p_t(x|s)$.

A general stream-level CFM loss for learning $u_t(x)$ is then

$$\mathcal{L}_{sCFM}(\theta) = \mathbb{E}_{t,s}\|v_t^\theta(s_t) - u_t(x|s)\|^2 = \mathbb{E}_{t,s}\|v_t^\theta(s_t) - \dot{s}_t\|^2$$

where the integration over $s$ is with respect to the marginal distribution of $s$ induced by $\pi(x_0, x_1)$ and $p_s(\cdot|x_0, x_1)$. As in previous research such as Lipman et al. (2023); Tong et al. (2024), we can show that the gradient of $\mathcal{L}_{sCFM}$ equals that of $\mathcal{L}_{FM}$ with details of the proof in Appendix J.2. However, stream-level CFM can be justified from a Bayesian decision-theoretic perspective without gradient matching. For more details, see Appendix A.

## 3.2. Choice of the Stream Model

Next, we specify the conditional model for the stream given the endpoints $p_s(\cdot|x_0, x_1)$. This model should emit streams differentiable with respect to time, with readily available velocity (either analytically or easily computable). Previous methods such as optimal transport (OT) conditional path/ linear interpolant (Liu et al., 2023; Lipman et al., 2023; Tong et al., 2024) achieve high sampling efficiency but can provide rather poor coverage of the $(t, x)$ space, resulting in extensive extrapolation of the estimated vector field $v_t^\theta(x)$. Furthermore, Albergo et al. (2023) demonstrated that the stochastic interpolant suppresses spurious intermediate features, thereby smoothing both the path and the vector field. Consequently, introducing stochasticity to the path simplifies the estimation and accelerates sample generation. Thus, it is desirable to consider stochastic models for streams that ensure smoothness while allowing streams to diverge and provide more spread-out coverage of the $(t, x)$ space.

To preserve the simulation-free nature of CFM, we consider models where the joint distribution of the stream and its velocity is available in closed form. In particular, we model the streams using Gaussian processes (GPs). A desirable property of a GP is that its velocity is also a GP, with mean and covariance directly derived from the original GP (Rasmussen & Williams, 2005). This enables efficient joint sampling of $(s_t, \dot{s}_t)$ given observations from a GP in stream-level CFM training. By adjusting covariance kernels for the joint GP, one can fine-tune the variance level to control the level of regularization, thereby further improving the estimation of the marginal vector field $u_t(x)$ (Section 4.1). The prior path constraints can also be incorporated into the kernel design. Additionally, a GP conditioning on the event that the stream passes through a finite number of intermediate locations between two endpoints again leads to a GP with analytic mean and covariance kernel (Section 4.2). This is particularly useful for incorporating multiple correlated observations.

Specifically, given $M$ time points $t = (t_1, t_2, \ldots, t_M)$ with $t_1 = 0$ and $t_M = 1$, we let $s_t = (s_{t_1}, s_{t_2}, \ldots, s_{t_M})$, and consider a more general conditional model for $p_s(\cdot \mid s_t =$

**Algorithm 1** Gaussian Process Conditional Flow Matching (GP-CFM)

---

**Input:** observation distribution $\pi(\boldsymbol{x}_{\text{obs}})$, initial network $v^\theta$, and a GP defining the conditional distribution $(s_t, \dot{s}_t) \mid \boldsymbol{s_t} = \boldsymbol{x}_{\text{obs}} \sim \mathcal{N}(\tilde{\boldsymbol{\mu}}_t, \tilde{\Sigma}_t)$, for $t \in [0, 1]$.
**Output:** fitted vector field $v_t^\theta(x)$.
**while** Training **do**
    $\boldsymbol{x}_{\text{obs}} \sim \pi(\boldsymbol{x}_{\text{obs}})$
    $t \sim U(0, 1)$
    $(s_t, \dot{s}_t) \mid \boldsymbol{s_t} = \boldsymbol{x}_{\text{obs}} \sim \mathcal{N}(\tilde{\boldsymbol{\mu}}_t, \tilde{\Sigma}_t)$
    $\mathcal{L}_{\text{sCFM}}(\theta) \leftarrow \|v_t^\theta(s_t) - \dot{s}_t\|^2$
    $\theta \leftarrow \text{update}\,(\theta, \nabla_\theta \mathcal{L}_{\text{sCFM}}(\theta))$
**end while**

---

$\boldsymbol{x}_{obs}$), where $\boldsymbol{x}_{obs} = (x_{t_1}, x_{t_2}, \ldots, x_{t_M})$ are a set of "observed values" that we require the statistic process $\boldsymbol{s}$ to pass through at time $(t_1, t_2, \ldots, t_M)$. Note that this contains the special case of conditioning on two endpoints (i.e., $M = 2$) described in Section 3.1. We consider a more general construction for $M \geq 2$ because later we will use this to incorporate multiple correlated observations (such as time series or other measurements from the same subject).

We construct a conditional GP for $\boldsymbol{s}$ that satisfies the boundary conditions, with a differentiable mean function $m$ and covariance kernel $k_{11}$. Since the derivative of a GP is also a GP, the joint distribution of $\boldsymbol{s}$ and the corresponding velocity process $\dot{\boldsymbol{s}} := \{\dot{s}_t : t \in [0, 1]\}$ given $\boldsymbol{s_t}$ is also a GP, with the mean function for $\dot{\boldsymbol{s}}$ being $\dot{m}(t) = dm(t)/dt$ and kernels defined by derivatives of $k_{11}$ (Rasmussen & Williams, 2005). To facilitate the construction of this GP, we consider an auxiliary GP on $\boldsymbol{s}$ with a differentiable mean function $\xi$ and covariance kernel $c_{11}$. Using the property that the conditional distribution of Gaussian remains Gaussian, we can obtain a joint GP model on $(\boldsymbol{s}, \dot{\boldsymbol{s}}) \mid \boldsymbol{s_t}$, which satisfies the boundary conditions. For computational efficiency and ease of implementation, we assume independence of the GP across dimensions of $\boldsymbol{s}$. Notably, while we are modeling streams conditionally given $\boldsymbol{s}$ as a GP, the marginal (i.e., unconditional) distribution of $\boldsymbol{s}$ at all time points is allowed to be non-Gaussian, which is necessary for satisfying the boundary condition and for the needed flexibility to model complex distributions. The detailed derivation can be found in Appendix C, and the training algorithm for GP-CFM is summarized in Algorithm 1.

Several conditional probability paths considered in previous works are special cases of the general GP representation. For example, if we set $m(t) = tx_1 + (1 - t)x_0$ (therefore, $\dot{m}(t) = x_1 - x_0$) and $k_{11}(t, t') = \sigma^2 \boldsymbol{I}_d$, the path reduces to the OT conditional path used in I-CFM with constant variance (Tong et al., 2024). The I-CFM path can also be induced by conditional GP construction (Appendix C) using a linear kernel for $c_{11}$, with more details in Appendix D. In

the following, we set $\xi(t) = 0$ and use squared exponential (SE) kernel for $c_{11}$ for each dimension (may be with additional terms such as in Figure 2). The details of SE kernel can be found in Appendix E.

Probability paths with time-varying variance, such as those considered in Song & Ermon (2019); Ho et al. (2020); Lipman et al. (2023), also motivate the adoption of non-stationary GPs whose covariance kernel could vary over $t$. For example, to encourage samples that display larger deviation from those in the training set (and hence more regularization), one could consider using a kernel producing larger variance as $t$ approaches the end with finite training samples (Figures 2 and 8). Moreover, because the GP model for $\boldsymbol{s}$ is specified given the two endpoints, both its mean and covariance kernel can be specified as functions of $(x_0, x_1)$. For example, if $x_1$ is an outlier of the training data, e.g., from a tail region of $q_1$, then one may incorporate a more variable covariance kernel for $p_{\boldsymbol{s}}(\cdot \mid x_0, x_1)$ to account for the uncertainty in the optimal transport path from $x_0$ to $x_1$.

# 4. Numerical Experiments

In this section, we demonstrate the benefits of GP stream models through several simulation examples. Specifically, we show that using GP stream models can improve the generated sample quality at a moderate cost of training time, by appropriately specifying the GP prior variance to reduce the sampling variance of the estimated vector field. Moreover, the GP stream model makes it easy to integrate multiple correlated observations along the time scale.

## 4.1. Adjusting GP Variance to Improve Sample Quality

We first show that one can improve the estimation of $u_t(x)$ by incorporating additional stochasticity in the sampling of streams with appropriate GP kernels. A similar observation has been made by Albergo et al. (2023), who also explained that this is because the stochastic interpolant can smooth the conditional probability path and suppress spurious intermediate modes. This applies to our GP-stream algorithm as well. See Appendix F for an illustration of the smoother conditional probability paths produced by our approach.

We have found it helpful to understand this phenomenon from the perspective of *variance reduction* in the estimation of the marginal vector field. As illustrated in Figure 1A, for estimating 2-Gaussian mixtures from standard Gaussian noise, the straight conditional stream used in I-CFM covers a relatively narrow region (gray). For points outside the searching region, there are no "data" and the neural network $v_t^\theta(x)$ must be extrapolated during sample generation. This is a form of overfitting, which causes highly variable estimates of the vector field in the extrapolated regions and can lead to potential "leaky" or outlying samples that are far

from the training observations.

In constructing the GP streams, we condition on the endpoints but expand the coverage region (red) by tweaking the kernel function (e.g., decrease the SE bandwidth in this case). This provides a layer of regularization that protects against extrapolation errors. As illustrated in Equation 1, the OT strategy for endpoints coupling (Tong et al., 2024) can be complementary to our GP-stream method to enhance performance. Therefore, we train the CNFs via four algorithms, i.e., I-CFM, GP-I-CFM, OT-CFM and GP-OT-CFM (OT for endpoints coupling and GP for stream model), 100 times using a 2-hidden layer multi-layer perceptron (MLP) with 100 training samples at $t = 1$, and calculate 2-Wasserstein (W2) distance between generated and test samples. For fair comparisons, we set $\sigma = 0$ for linear interpolant (I-CFM and OT-CFM), and use noise-free GP streams (GP-I-CFM and GP-OT-CFM). The results are summarized in Table 1. Empirically, the models trained by GP-stream CFM have smaller W2 distance than the corresponding linear interpolant (i.e., GP-I-CFM vs. I-CFM and GP-OT-CFM vs. OT-CFM), and the model trained by combining two strategies (GP-OT-CFM, OT for coupling and GP for stream) performs best. We further generate 1000 samples and streams for I-CFM and GP-I-CFM with the largest W2 distance in Figure 1B, starting with the same points from standard Gaussian. In this example, several outliers are generated from I-CFM.

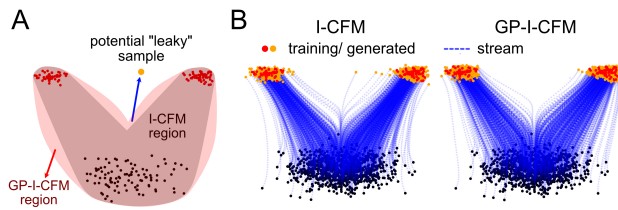

*Figure 1.* **GP streams reduce extrapolation by expanding coverage area**. Generate samples of 2-Gaussian mixture from the standard Gaussian. Training observations are shown in red, generated samples in orange, and noise source samples in black. **A**. FM with straight conditional stream (e.g., I-CFM) may generate "leaky" or outlier samples due to extrapolation errors. The FM method with GP conditional stream has a broader coverage area. **B**. We train models with I-CFM and GP-I-CFM 100 times and calculate 2-Wasserstein (W2) distance. Among these 100 trained models, generate 1000 samples (orange) and streams (blue) for I-CFM and GP-I-CFM with largest W2 distance (worst case).

We can further modify the GP variance function over time to efficiently improve sample quality. Here, we consider the task of estimating and sampling a 2-Gaussian mixture from the standard Gaussian, with 100 training samples at $t = 1$. For constant noise, diagonal white noise is added to perturb stream locations while retaining the SE kernel. For varying noise, we add a non-stationary dot

*Table 1.* **Comparison of linear and GP streams** Consider generating 2-Gaussian target from standard Gaussian. Here, we train models with I-CFM, GP-I-CFM, OT-CFM and GP-OT-CFM 100 times and calculate 2-Wasserstein (W2) distance between generated and test samples. Results of 100 seeds are summarized by mean and standard error.

| MODELS | MEAN | SE |
|---|---|---|
| I-CFM | 1.54 | 0.08 |
| GP-I-CFM | 1.51 | 0.08 |
| OT-CFM | 1.43 | 0.05 |
| GP-OT-CFM | 1.35 | 0.05 |

product kernel to the SE kernel. Specifically, denote the kernel for auxiliary GP on $s$ in dimension $i$ as $c_{11}^i$, for $i = 1, \ldots, d$. Let $c_{11}^i(t, t') = c_{11}^{\text{SE}}(t, t') + \alpha tt'$ for increasing variance and $c_{11}^i(t, t') = c_{11}^{\text{SE}}(t, t') + \alpha(t-1)(t'-1)$ for decreasing variance, where $\{t, t'\} \in [0, 1]$ and $c_{11}^{\text{SE}}(t, t') = \sigma^2 \exp\left(-\frac{(t-t')^2}{2l^2}\right)$. (See Appendix C for additional details.) Some examples of the streams connecting two endpoints under different variance schemes are shown in Figure 2. We train the models 100 times and calculate the 2-Wasserstein (W2) distance between generated and test samples, and the results are summarized in Table 2. In this example, with infinite samples at $t = 0$ but 100 samples at $t = 1$, injecting noise at $t = 0$ worsens estimation. However, when approaching the target distribution ($t = 1$), adding noise can improve estimation with small samples (100). The noise perturbs the limited data, encouraging broader exploration and adding regularization to reduce estimation error. In addition to using a standard Gaussian source, we further consider the transformation between two 2-Gaussian mixtures with finite samples (100) at both ends. Results are shown in Appendix G. In this scenario, injecting noise near either endpoint improves estimation.

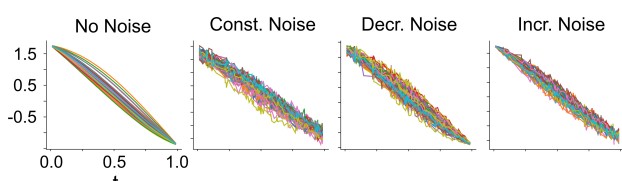

*Figure 2.* **Change variance over time by tweaking the covariance kernel**. Examples of conditional stream between two points, under different variance change scheme.

## 4.2. Incorporating Multiple Correlated Training Observations

Besides generally improving FM-based fitting of a marginal vector field, an additional benefit of GP streams is that they

*Table 2.* **Comparison of different variance schemes of GP-I-CFM** Reconsider the generation of 2-Gaussian target from standard Gaussian. We train models under each variance scheme 100 times and calculate 2-Wasserstein (W2) distance for each. The results of 100 seeds are summarized by mean and standard error.

| VARIANCE SCHEME | MEAN | SE |
|---|---|---|
| NO NOISE | 0.269 | 0.013 |
| CONSTANT NOISE | 0.305 | 0.011 |
| DECREASING NOISE | 0.329 | 0.013 |
| INCREASING NOISE | 0.243 | 0.012 |

enable the flexible inclusion of multiple correlated observations in the training data, such as in time series. Correlations between training observations allow information sharing and can improve estimation at each time point.

We first illustrate the main idea through a toy example. Consider 100 paired observations and place the two observations in each pair at $t = 0.5$ and $t = 1$, respectively (Figure 3 A) while $t = 0$ corresponds to the standard Gaussian source. Here, we show the generated samples (at $t = 0.5$ and $t = 1$) and the corresponding streams for GP-I-CFM and I-CFM. Again, 2-hidden layer MLP is used in this case. The I-CFM strategy employs two separate models with I-CFM algorithms (Figure 3B), whereas GP-I-CFM offers a single unifying model for all observations, resulting in a smooth stream across all time points (Figure 3C).

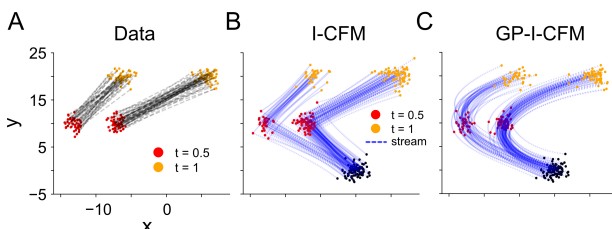

*Figure 3.* **GP streams accommodate correlated points flexibly**. **A**. Paired data with observations on t = 0.5 (red) and t = 1 (orange). **B**. The generated samples (red for t = 0.5 and orange for t = 1) and streams (blue) for I-CFMs. The I-CFMs contain two separate models trained by I-CFM, t = 0 (standard Gaussian noise) to t = 0.5 and t = 0.5 to t = 1. **C**. The generated samples for GP-I-CFM.

In some cases, the GP streams may not be well separated, and thus may confuse the training of the vector field at crossing points. In Figure 4, we show a time series dataset over 3 time points, where training data at $t = 0$ and $t = 1$ are on one horizontal side while points at $t = 0.5$ are on the opposite side (Figure 4A). Therefore, these streams have two crossing regions (marked with blue boxes in Figure 4A), where the training of the vector field is deteriorated when simply using the GP-I-CFM (Figure 4B). An

easy solution is to further condition the neural net $v_t^\theta(x)$ on covariate (subject label) $c$, so that the optimization objective is $\mathcal{L}_{\text{cCFM}} = \mathbb{E}_{t\sim U(0,1), s\sim q(s|c)}\|v_t^\theta(s_t, c) - \dot{s}_t\|^2$, where $q(s \mid c)$ represents the distribution of $s$ given $c$. The covariate-dependent FM (guided-FM) algorithm has been proposed in Isobe et al. (2024); Zheng et al. (2023), and the validity of approximating the covariate-dependent vector field using the above optimization objective in our stream-level CFM is shown in the Appendix J.3. In this example, similar subjects have close starting points at $t = 0$, and we let $c = x_0$. By conditioning on $c$ (covariate model), the neural net is separated for different subjects, and hence the training of the vector field will not be confused (Figure 4C).

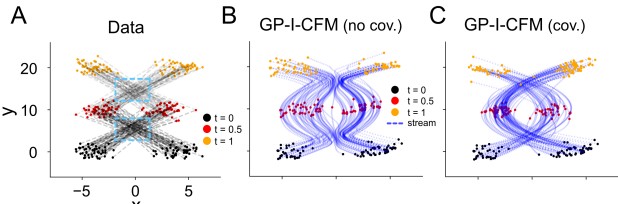

*Figure 4.* **Further conditioning on the starting points helps with stream generation**. **A**. Paired data with observations on three time points: t = 0 (black), t = 0.5 (red) and t = 1 (orange). The two stream cross regions are marked with light blue square. **B**. The generated samples and streams for GP-I-CFM (without covariate), where the initial points at $t = 0$ are generated from noise using a separate I-CFM. **C**. The generated samples and streams for GP-I-CFM with covariate using the same starting points, where the neural network is further conditioning on data at $t = 0$.

## 5. Applications

We apply our GP-based CFM methods to two hand-written image datasets (MNIST and HWD+), CIFAR-10 dataset, and a mouse brain local field potential (LFP) dataset to illustrate how GP-based algorithms 1) reduce sampling variance (MNIST and CIFAR-10) and 2) flexibly incorporate multiple correlated observations and generate smooth transformations across different time points (HWD+ and LFP dataset). The reported running times for the experiments are obtained on a server configured with 2 CPUs, 24 GB RAM, and 2 RTXA5000 GPUs.

### 5.1. Variance Reduction

We explore the empirical benefits of variance reduction using FM with GP conditional streams on MNIST (Deng, 2012) and CIFAR-10 (Krizhevsky, 2009) databases. Here, we consider four algorithms in the MNIST application: two linear stream models (I-CFM, OT-CFM) and two GP stream models (GP-I-CFM, GP-OT-CFM). The I-CFM and GP-I-CFM are implemented for the CIFAR-10 example.

In the MNIST application, we set $\sigma = 0$ for linear stream models and use noise-free GP stream models for fair comparisons. U-Nets (Ronneberger et al., 2015; Nichol & Dhariwal, 2021) with 32 channels and 1 residual block are used for all models. It takes around 50s, 51s, 52s, and 53s for I-CFM, OT-I-CFM, GP-I-CFM, and GP-OT-CFM to pass through all training dataset once for model training. To evaluate how much the GP stream-level CFM can further improve the estimation, we train each algorithm 100 times, and calculate the kernel inception distance (KID) (Bińkowski et al., 2018) and Fréchet inception distance (FID) (Heusel et al., 2017) for each. The histograms in Figure 5 show the distribution of these 100 KIDs and FIDs, with results summarized in Table 3. According to KID and FID, the independent sampling algorithms (I-algorithms) are comparable to optimal transport sampling algorithms (OT-algorithms). However, algorithms using GP conditional stream exhibit lower standard error and fewer extreme values for KID and FID, thereby reducing the occurrence of outlier samples (as in Figure 1).

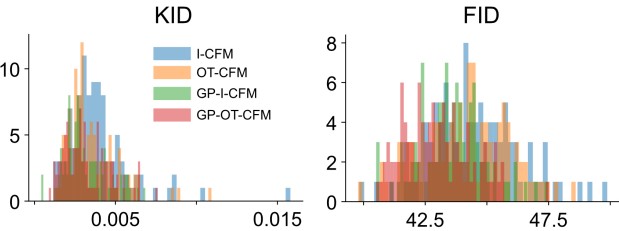

*Figure 5.* **Application to MNIST dataset**. We compare the performance of four algorithms (I-CFM, OT-CFM, GP-I-CFM and GP-OT-CFM) on fitting MNIST dataset. We fit the models 100 times for each, and evaluate the quality of the samples by KID and FID. The figures above show the historams of KID and FID.

Besides the MNIST dataset, to evaluate the performance in the high-dimensional image setting, we perform an experiment on unconditional CIFAR-10 generation (Krizhevsky, 2009) from a standard Gaussian source. We use a similar setup to that of Tong et al. (2024), such as time-dependent U-Net (Ronneberger et al., 2015; Nichol & Dhariwal, 2021) with 128 channels, a learning rate of $2 \times 10^{-4}$, clipping gradient norm to 1.0 and exponential moving average with a decay of 0.9999. Again, four algorithms (I-CFM, OT-CFM, GP-I-CFM, and GP-OT-CFM) are implemented. We add diagonal white noise $10^{-6}$ into GP-stream models to prevent a potential singular GP covariance matrix, and set $\sigma = 10^{-3}$ in linear interpolations for fair comparisons. The models are trained for 400,000 epochs, with a batch size of 128. The linear interpolation (I-models) runs around 3.6 iterations per second, while GP-stream (GP-models) runs around 3.0 iterations per second. Figure 9A in Appendix H shows 64 generated images from four trained models, using a DOPRI5 adaptive solver. Visually, images generated by GP-stream (e.g., GP-I-CFM) are generally sharper

and exhibit more details compared to those generated by the linear interpolant (I-CFM). The mean (with standard error) Fréchet inception distance (FID) (Heusel et al., 2017), calculated by the clean-fid library (Parmar et al., 2022) with 50,000 samples and the running time to generate 10 images 20 times using the DOPRI5 solver, is reported in Table 4.

In terms of FID, GP-I-CFM is the best and significantly improves the I-CFM. In this case, the OT-CFM is comparable to I-CFM, as observed in the MNIST application (Figure 5 and Table 3). It may suggest that the benefit of OT-CFM is less significant with increasing dimension, since in the high-dimensional case, minibatch OT approximation is poor for true OT and further adopting the GP stream strategy (GP-OT-CFM) does not remedy the issue. In terms of sample generation time, using GP stream or OT coupling strategy leads to a more computationally efficient model. However, combining these two strategies does not improve the efficiency of sample generation in this case.

### 5.2. Multiple Training Observations

Finally, we demonstrate how our GP stream-level CFM can flexibly incorporate correlated observations (between two endpoints at $t = 0$ and $t = 1$) into a single model and provide a smooth transformation across different time points, using the HWD+ dataset (Beaulac & Rosenthal, 2022) and LFP dataset (Steinmetz et al., 2019). The HWD+ example concerns transformations on artificial time, and the LFP dataset is time series data from the mouse brain. Here, we show results for the HWD+ dataset; refer to Appendix I for the LFP application.

The HWD+ dataset contains images of handwritten digits along with writer IDs and characteristics, which are not available in the MNIST dataset used in Section 5.1. Here, we consider the task of transforming from "0" (at $t = 0$) to "8" (at $t = 0.5$), and then to "6" (at $t = 1$). The intermediate image, "8", is placed at $t = 0.5$ (artificial time) for "symmetric" transformations. All three images have the same number of samples, totaling 1,358 samples (1,086 for training and 272 for testing) from 97 subjects. The U-Nets with 32 channels and 1 residual block are used. Both models with and without covariates (using starting images, as in Figure 4C) are considered. Each model is trained both by I-CFM and GP-I-CFM. The I-CFM transformation contains two separate models trained by I-CFM ("0" to "8" and "8" to "6"). Noise-free GP-I-CFM and I-CFM with $\sigma = 0$ are used for fair comparisons. In each training iteration, we randomly select samples within each writer to preserve the grouping structure of data. The runtime for all algorithms (I-CFM, GP-I-CFM and corresponding labeled versions) is similar, which takes 0.74s for passing all training data once. However, since I-CFMs fit two separate models, the running time is doubled.

*Table 3.* **Comparison of different algorithms for MNIST dataset** Train models 100 times using different algorithms. Calculate the mean and standard error for KID and FID.

| | | I-CFM | OT-CFM | GP-I-CFM | GP-OT-CFM |
|---|---|---|---|---|---|
| KID | MEAN | 0.0040 | 0.0036 | 0.0032 | 0.0031 |
| | SE | 0.0002 | 0.0002 | 0.0001 | 0.0001 |
| FID | MEAN | 44.50 | 44.21 | 43.55 | 42.99 |
| | SE | 0.18 | 0.17 | 0.13 | 0.14 |

*Table 4.* **Comparison of different algorithms for CIFAR-10 dataset** We fit models by I-CFM, OT-CFM, GP-I-CFM and GP-OT-CFM. For each trained model, we 1) calculate FID using 50000 samples 20 times and 2) generate 10 images 20 times. The means and standard errors of each are summarized as follows.

| MODELS | FID | | SAMPLE GEN. TIME (S) | |
|---|---|---|---|---|
| | MEAN | SE | MEAN | SE |
| I-CFM | 3.75 | 0.006 | 1.30 | 0.01 |
| OT-CFM | 3.74 | 0.009 | 1.07 | 0.02 |
| GP-I-CFM | 3.62 | 0.008 | 1.27 | 0.02 |
| GP-OT-CFM | 3.75 | 0.009 | 1.20 | 0.01 |

The traces for 10 generated samples from each algorithm are shown in Figure 9B in Appendix H, where the starting images ('0' in the first rows) are generated by an I-CFM from standard Gaussian noise. Visually, the GP-based algorithms generate higher quality images and smoother transformations compared to algorithms using linear conditional stream (I-CFM), highlighting the benefit of including correlations across different time points. Additionally, the transformation generally looks smoother when the CFM training is further conditioned on the starting images.

We then quantify the performance of different algorithms by calculating the FID for "0", "8" and "6", and plot them over time for each (Figure 6). For all FIDs, the GP-based algorithms (green & red) outperform their straight connection (I-) counterparts (blue & orange) , especially for the FID for "8" at $t = 0.5$ and the FID to "6" at $t = 1$. This also holds for the FID for "0", as the GP-based algorithms are unified and the information is shared across all time points. This aligns with the observation by Albergo et al. (2024) that jointly learning multiple distributions better preserves the original image's characteristics during translation. However, for the I-algorithms, the conditional version (orange) performs worse than the unconditional one (blue), as conditioning on the starting images makes the stream more separated, requiring more data to achieve comparable performance. In contrast, the data in GP-based algorithms is more efficiently utilized, as correlations across time points for the same subject are integrated into one model. Therefore,

explicitly accounting for the grouping effect by conditioning on starting images (red) further improves performance.

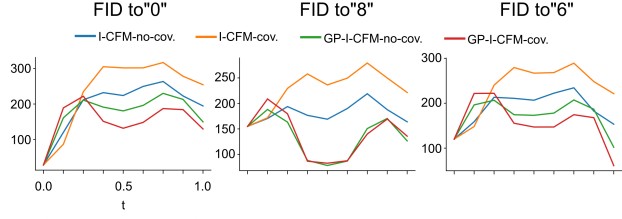

*Figure 6.* **Application to HWD+ dataset**. We fit models for transforming "0" to "8" and then to "6". Both covariate and non-covariate (on starting images) models are considered, and each model is fitted by both I-CFM and GP-I-CFM. The I-CFM transformation consists of two separate models trained by I-CFM ("0" to "8" and "8" to "6"). The figures above show corresponding FID to "0", "8" and "6" for these four trained models over time.

## 6. Conclusion

We extend CFM algorithms using latent variable modeling. In particular, we adopt GP models on the latent streams and propose a class of CFM algorithms based on sampling along the streams. Our GP-stream algorithm preserves the simulation-free feature of CFM training by exploiting distributional properties of GPs. Not only can our GP-based stream-level CFM reduce the variance in the estimated vector field thereby improving the sample quality, but it allows easy integration of multiple correlated observations to achieve borrowing of strength. The GP-CFM is complementary to and can be combined with modeling on the coupling of endpoints (e.g., OT-CFM).

A potential drawback of GP-CFM is the Monte Carlo error induced in the vector field estimation. This error can be mitigated through techniques such as importance sampling over time $t$, as in Song et al. (2021a) and antithetic sampling of GP trajectories (Botev & Ridder, 2017).

## Acknowledgments

This research was supported by NIGMS grant R01-GM135440 and NSF grant EEC-2133504.

## Impact Statement

This paper presents work whose goal is to advance the field of deep generative models. The proposed method can potentially help generate samples more efficiently with higher quality in many (scientific) fields. There may be some potential societal consequences of our work, none of which we feel must be specifically highlighted here.

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

## A. Bayesian Decision Theoretic Perspective on (stream-level) Flow Matching Training

It is well-known in Bayesian decision theory (Berger, 1985) that under squared error loss, the Bayesian estimator, which minimizes both the posterior expected loss (which conditions on the data and integrates out the parameters) and the marginal loss (which integrates out both the parameters and the data), is exactly the posterior expectation of that parameter. This implies immediately that if one considers the conditional vector field $u_t(x|z)$ as the target of "estimation", and the corresponding "data" being the event that $x_t = x$, i.e., that the path goes through $x$ at time $t$, then the corresponding Bayes estimate for $u_t(x|z)$ will be exactly the marginal vector field $u_t(x)$, as it is now the "posterior mean" of $u_t(x|z)$. We emphasize again that here the "data" differs from the actual training and the generated noise observations, which in fact help form the "prior" distribution. Therefore, the FM objective ($\mathcal{L}_{\text{FM}}$) defined in Section 2 provides a reasonable approximation to $u_t(x)$.

In stream-level FM algorithm, because the (population-level) minimizer for the sCFM loss is $u_t(x)$, minimizing the sCFM loss provides a reasonable estimate for the marginal vector field $u_t(x)$. To see this, rewrite the sCFM loss by the law of iterated expectation as

$$\mathcal{L}_{\text{sCFM}}(\theta) = \mathbb{E}_t \mathbb{E}_{\boldsymbol{s}} \left( \|v_t^\theta(s_t) - \dot{s}_t\|^2 | t \right).$$

The inner expectation can be further written in terms of another iterated expectation:

$$\mathbb{E}_{\boldsymbol{s}} \left( \|v_t^\theta(s_t) - \dot{s}_t\|^2 | t \right) = \mathbb{E}_{s_t} \mathbb{E}_{\boldsymbol{s}} \left( \|v_t^\theta(s_t) - \dot{s}_t\|^2 | t, s_t \right).$$

For any $x$, $\mathbb{E}_{\boldsymbol{s}} \left( \|v_t^\theta(s_t) - \dot{s}_t\|^2 | t, s_t = x \right) = \mathbb{E}_{\boldsymbol{s}} \left( \|v_t^\theta(x) - \dot{s}_t\|^2 | t, s_t = x \right)$, whose minimizer is the conditional expectation of $\dot{s}_t$ given $s_t = x$, which is exactly $u_t(x)$. Hence, one can estimate $u_t(x)$ by minimizing $\mathcal{L}_{\text{sCFM}}(\theta)$. This justifies training $u_t(x)$ through the sCFM loss without regard to any specific optimization strategy.

## B. Discussion on Per-stream Perspective on Flow Matching

It is helpful to recognize the relationship between the per-stream vector field and the conditional vector field given one or both endpoints introduced previously in the literature. Specifically, the per-sample vector field in Lipman et al. (2023) corresponds to marginalizing out $\boldsymbol{s}$ given the end point $x_1$, that is, $u_t(x \mid x_1) = \mathbb{E}\left(u_t(x \mid \boldsymbol{s}) \mid s_t = x, s_1 = x_1\right)$. Similarly, the conditional vector field of Tong et al. (2024) corresponds to marginalizing out $\boldsymbol{s}$ given both $x_0$ and $x_1$, that is $u_t(x \mid x_0, x_1) = \mathbb{E}\left(u_t(x \mid \boldsymbol{s}) \mid s_t = x, s_0 = x_0, s_1 = x_1\right)$. Furthermore, when $p_{\boldsymbol{s}}(\cdot \mid x_0, x_1)$ is simply a unit-point mass (Dirac) concentrated on the optimal transport (OT) path, i.e., a straight line that connects two endpoints $x_0$ and $x_1$, then $u_t(x \mid \boldsymbol{s}) = u_t(x \mid x_1) = u_t(x \mid x_0, x_1)$ for all $(\boldsymbol{s}, t, x)$ tuples that satisfy $s_0 = x_0, s_1 = x_1, s_t = x$. Intuitively, when the stream connecting two ends is unique, conditioning on the two ends is equivalent to conditioning on the corresponding stream $\boldsymbol{s}$. In this case, our stream-level FM algorithm (Section 3.2) coincides with those previous algorithms. More generally, however, this equivalence does not hold when $p_{\boldsymbol{s}}(\cdot \mid x_0, x_1)$ is non-degenerate.

The per-stream view affords additional modeling flexibility and alleviates the practitioners from the burden of directly sampling from the conditional probability paths given one (Lipman et al., 2023) or both endpoints (Tong et al., 2024). While the per-stream vector field induces a degenerate unit-point mass conditional probability path, we will attain non-degenerate marginal and conditional probability paths that satisfy the boundary conditions after marginalizing out the streams. Sampling the streams in essence provides a data-augmented Monte Carlo alternative to sampling directly from the conditional probability paths, which can then allow estimation of the marginal vector field $u_t(x)$ when direct sampling from the conditional probability path is challenging. Additionally, as we will demonstrate later, by approaching FM at the stream level, one could more readily incorporate prior knowledge or other external features into the design of the stream distribution $p_{\boldsymbol{s}}(\cdot \mid x_0, x_1)$.

## C. Derivation of joint conditional mean and covariance

For computational efficiency and ease of implementation, we assume independent GPs across dimensions and present the derivation dimension-wise throughout the Appendices. We use $s_t^i$ to denote the location of stream $\boldsymbol{s}$ at time $t$ in dimension $i$, for $i = 1, \ldots, d$. Suppose each dimension of stream $\boldsymbol{s}$ follows a Gaussian process with a differentiable mean function $\xi^i$ and covariance kernel $c_{11}^i$. Then, the joint distribution of $s_{t_1, \ldots, t_g}^i = (s_{t_1}^i, \ldots, s_{t_g}^i)'$ and $\dot{s}_{t_1, \ldots, t_g}^i = (\dot{s}_{t_1}^i, \ldots, \dot{s}_{t_g}^i)'$ at $g$ time points is

$$\begin{pmatrix} s^i_{t_1,\ldots,t_g} \\ \dot{s}^i_{t_1,\ldots,t_g} \end{pmatrix} \sim \mathcal{N}\left( \begin{pmatrix} \xi^i_{t_1,\ldots,t_g} \\ \dot{\xi}^i_{t_1,\ldots,t_g} \end{pmatrix}, \begin{pmatrix} \Sigma^i_{11} & \Sigma^i_{12} \\ \Sigma^{i}_{12}{}^\mathsf{T} & \Sigma^i_{22} \end{pmatrix} \right), \tag{3}$$

where $\xi^i_t = \xi^i(t)$, $\dot{\xi}^i_t = \mathrm{d}\xi^i_t/\mathrm{d}t$, $\xi^i_{t_1,\ldots,t_g} = (\xi^i_{t_1},\ldots,\xi^i_{t_g})'$, $\dot{\xi}^i_{t_1,\ldots,t_g} = (\dot{\xi}^i_{t_1},\ldots,\dot{\xi}^i_{t_g})'$ and covariance $\Sigma^i_{jl}$ is determined by kernel $c^i_{jl}$. The kernel function for the covariance between $s$ and $\dot{s}$ in dimension $i$ is $c^i_{12}(t,t') = \frac{\partial c^i_{11}(t,t')}{\partial t'}$, and the kernel defining covariance of $\dot{s}$ is $c^i_{22} = \frac{\partial^2 c^i_{11}(t,t')}{\partial t \partial t'}$ (Rasmussen & Williams (2005) Chapter 9.4). The conditional distribution of $(s,\dot{s})$ in dimension $i$ given $M$ observations $s^i_t = x^i_{\mathrm{obs}}$ is also a (bivariate) Gaussian process. In particular, for $t \in [0,1]$, let $\boldsymbol{\mu}^i_t = (\xi^i_t, \dot{\xi}^i_t)'$ and $\boldsymbol{\mu}^i_{\mathrm{obs}} = (\xi^i_{t_1},\ldots,\xi^i_{t_s})$, the joint distribution is

$$\left( s^i_t, \dot{s}^i_t, {\boldsymbol{x}^i_{\mathrm{obs}}}' \right)' \sim \mathcal{N}\left( \begin{pmatrix} \boldsymbol{\mu}^i_t \\ \boldsymbol{\mu}^i_{\mathrm{obs}} \end{pmatrix}, \begin{pmatrix} \Sigma^i_t & \Sigma^i_{t,\mathrm{obs}} \\ \Sigma^{i}{}^\mathsf{T}_{t,\mathrm{obs}} & \Sigma^i_{\mathrm{obs}} \end{pmatrix} \right),$$

where $\Sigma^i_t = \mathrm{Cov}(s^i_t, \dot{s}^i_t)$ and $\Sigma^i_{\mathrm{obs}} = \mathrm{Cov}(\boldsymbol{x}^i_{\mathrm{obs}})$. Accordingly, the conditional distribution $(s^i_t, \dot{s}^i_t)\,|\,\boldsymbol{x}^i_{\mathrm{obs}} \sim \mathcal{N}(\tilde{\boldsymbol{\mu}}^i_t, \tilde{\Sigma}^i_t)$, where $\tilde{\boldsymbol{\mu}}^i_t = \boldsymbol{\mu}^i_t + \Sigma^i_{t,\mathrm{obs}}{\Sigma^i_{\mathrm{obs}}}^{-1}(\boldsymbol{x}^i_{\mathrm{obs}} - \boldsymbol{\mu}^i_{\mathrm{obs}})$ and $\tilde{\Sigma}^i_t = \Sigma^i_t - \Sigma^i_{t,\mathrm{obs}}{\Sigma^i_{\mathrm{obs}}}^{-1}\Sigma^{i}{}^\mathsf{T}_{t,\mathrm{obs}}$.

## D. Optimal transport path from Conditional GP Construction

In this section, we show how to derive the path in I-CFM (Tong et al., 2024) from the conditional GP construction (Appendix C) using a linear kernel. Without loss of generality, we present the derivation of "noise-free" path with $\sigma^2 = 0$ (i.e., the rectified flow, Liu et al. (2023); Liu (2022)).

Let $\boldsymbol{x}^i_{\mathrm{obs}} = (x^i_0, x^i_1)'$, $\xi^i_t = \dot{\xi}^i_t = 0$ and $c^i_{11}(t,t') = \sigma^2_a + \sigma^2_b(t-1)(t'-1)$, such that

$$\Sigma^i_t = \begin{pmatrix} \sigma^2_a + \sigma^2_b(t-1)^2 & \sigma^2_b(t-1) \\ \sigma^2_b(t-1) & \sigma^2_b \end{pmatrix}, \qquad \Sigma^i_{t,\mathrm{obs}} = \begin{pmatrix} \sigma^2_a - \sigma^2_b(t-1) & \sigma^2_a \\ -\sigma^2_b & 0 \end{pmatrix},$$

$$\Sigma^i_{\mathrm{obs}} = \begin{pmatrix} \sigma^2_a + \sigma^2_b & \sigma^2_a \\ \sigma^2_a & \sigma^2_a \end{pmatrix}, \qquad\qquad {\Sigma^i_{\mathrm{obs}}}^{-1} = \frac{1}{\sigma^2_b}\begin{pmatrix} 1 & -1 \\ -1 & 1 + \frac{\sigma^2_b}{\sigma^2_a} \end{pmatrix}.$$

Therefore,

$$\tilde{\boldsymbol{\mu}}^i_t = \Sigma^i_{t,\mathrm{obs}}{\Sigma^i_{\mathrm{obs}}}^{-1}\begin{pmatrix} x^i_0 \\ x^i_1 \end{pmatrix} = \begin{pmatrix} 1-t & t \\ -1 & 1 \end{pmatrix}\begin{pmatrix} x^i_0 \\ x^i_1 \end{pmatrix} = \begin{pmatrix} (1-t)x^i_0 + tx^i_1 \\ x^i_1 - x^i_0 \end{pmatrix},$$

$$\tilde{\Sigma}^i_t = \Sigma^i_t - \Sigma^i_{t,\mathrm{obs}}{\Sigma^i_{\mathrm{obs}}}^{-1}\Sigma^{i}{}^\mathsf{T}_{t,\mathrm{obs}} = \boldsymbol{O}$$

## E. Covariance under Squared Exponential kernel

Throughout this paper, we adopted the squared exponential (SE) kernel, with the same hyper-parameters for each dimension. The kernel defining block covariance for $s$, $(s, \dot{s})$ and $\dot{s}$ in dimension $i$ from Equation 3 are as follows:

$$c^i_{11}(t,t') = \alpha \exp\left(-\frac{(t-t')^2}{2l^2}\right) \qquad c^i_{12}(t,t') = \frac{\alpha}{l^2}(t-t')\exp\left(-\frac{(t-t')^2}{2l^2}\right)$$

$$c^i_{21}(t,t') = -c^i_{12}(t,t') \qquad c^i_{22}(t,t') = \frac{\alpha}{l^4}\left[l^2 - (t-t')^2\right]\exp\left(-\frac{(t-t')^2}{2l^2}\right).$$

## F. GP stream produces a smoother probability path

The proposed GP-stream model can be considered as a more general framework than the stochastic interpolant (Albergo et al., 2023), which can produce a smoother path and suppress spurious intermediate modes. This also holds for our GP-stream model, and the smooth path makes it easier for vector field estimation and more computationally efficient numerical integration for sample generations. The running time for generating high-dimensional images, i.e., CIFAR-10, significantly shows the computational benefit (Table 4 in Section 5.1).

Here, we consider the transformation from a 2-Gaussian mixture to a 3-Gaussian mixture. The models trained by four algorithms are compared: I-CFM, OT-CFM, GP-I-CFM, GP-OT-CFM. The generated paths and samples are shown in Figure 7. Generally, either I-CFM or OT-CFM produces spurious modes, and the path is not smooth.

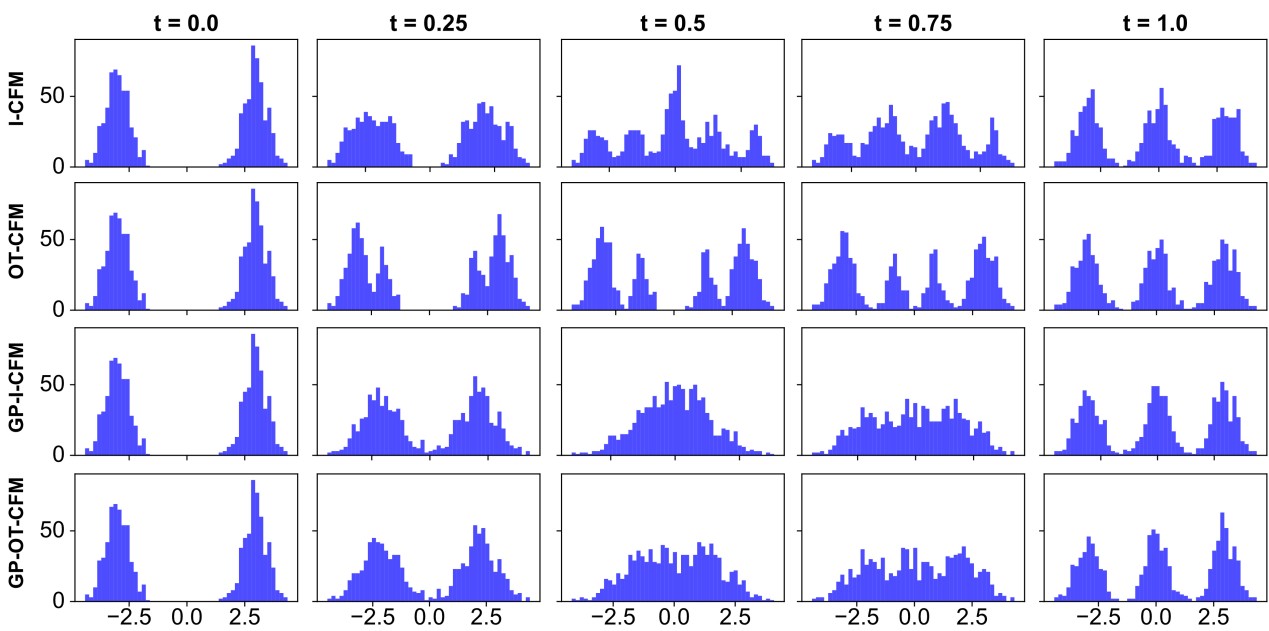

*Figure 7.* **GP streams provide smoother probability paths**. Here, we consider the transformation from a 2-Gaussian mixture to a 3-Gaussian mixture. We fit models by four algorithms: I-CFM, OT-I-CFM, GP-I-CFM and GP-OT-CFM. The generated samples at $t = 0, 0.25, 0.5, 0.75$ and $1$ for each model is visualize by histograms.

## G. A Supplementary Example for Variance Changing over Time

Here, instead of generating data from standard Gaussian noise, we consider 100 training (unpaired) samples from a 2-Gaussian to another 2-Gaussian (Figure 8A). The example streams connecting two points under different variance schemes are shown in Figure 8B, again using additional nugget noise for constant noise, and a dot product kernel for decreasing and increasing noise, as described in Section 4.1. We then fit 100 independent models and calculate the W2 distance between generated and test samples at $t = 1$. The results are summarized in Table 5. Now, since both ends have finite samples, injecting noise (a.k.a. adding regularization) at both ends helps.

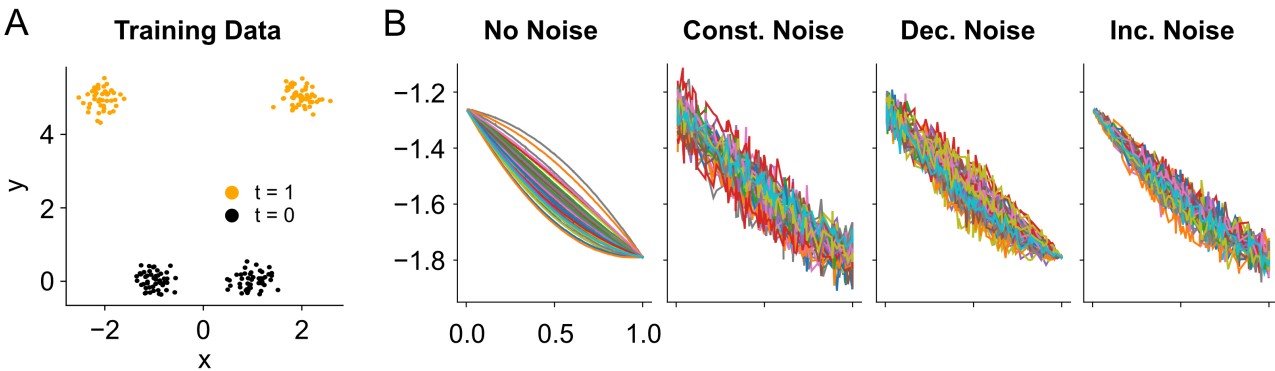

*Figure 8.* **Supplementary Example for Variance Change over Time**.**A**. The 100 observations in training data at $t = 0$ and $t = 1$. **B**. Examples of streams between two points, under different variance change scheme.

*Table 5.* **Comparison of different variance schemes of GP-I-CFM for 2-Gaussian to 2-Gaussian** Train models 100 times and calculate 2-Wasserstein (W2) distance between generated and test samples for each. The results of these 100 seeds are summarized by mean and standard error.

|  | NO NOISE | CONSTANT NOISE | DECREASING NOISE | INCREASING NOISE |
|---|---|---|---|---|
| MEAN | 0.681 | 0.350 | 0.413 | 0.411 |
| SE | 0.044 | 0.036 | 0.032 | 0.032 |

## H. Image Generations

In this section, we show generated sample images for the CIFAR-10 and HWD+ dataset.

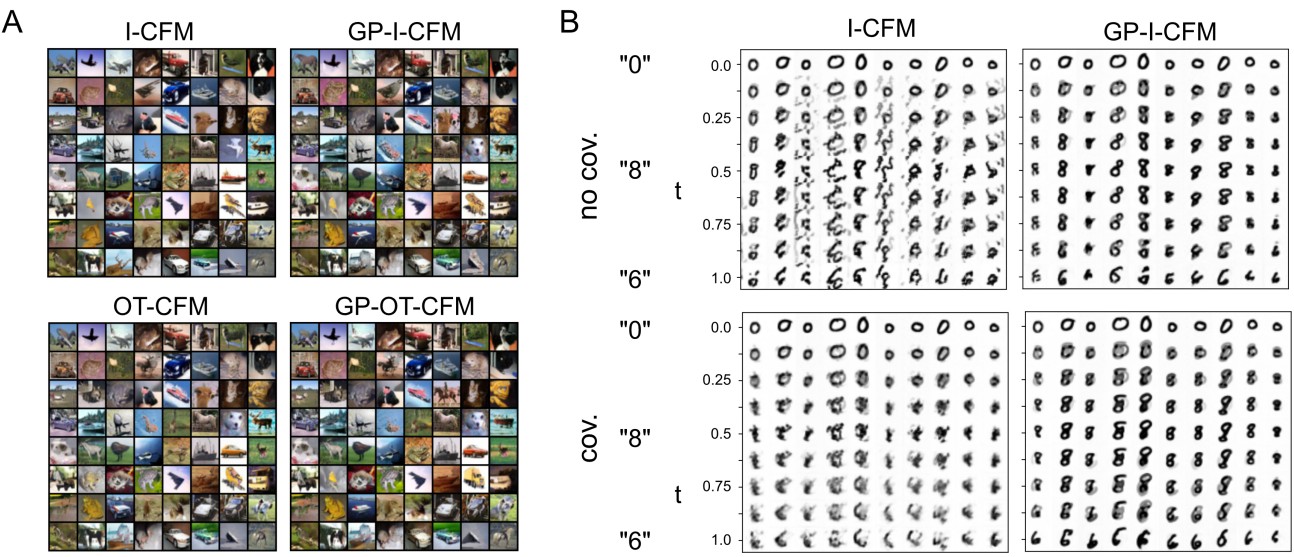

*Figure 9.* **Image Generation**. **A**. 64 generated CIFAR-10 samples for I-CFM, GP-I-CFM, OT-CFM and GP-OT-CFM, starting from the same standard Gaussian samples. **B**. 10 HWD+ sample traces for the four trained models. The starting images ("0"s in the first row) are generated by an I-CFM from standard Gaussian noise, and all four trained models use the same starting images.

## I. Application to LFP dataset

In this section, to illustrate the usage of proposed GP-CFM for time series data, we apply the labeled-GP-I-CFM to a session of local field potential (LFP) data from a mouse brain. In the LFP dataset, the neural activity across multiple brain regions is recorded when the mice perform a task on choosing the side with the highest contrast for visual gratings. The data contains 39 sessions from 10 mice, and each session contains multiple trials. Time bins for all measurements are 10 ms, starting 500 ms before stimulus onset. Here, we study LFP from stimulus onset to 500ms after stimulus, and hence each trial contains data from 50 time points. See Steinmetz et al. (2019) for more details on the LFP dataset.

Here, we choose recordings from a mouse in one session, where the trial is repeated 214 times. For each single trial, the data contains a time series from 7 brain regions. To illustrate the temporal smoothness over time in a visually significant way, we subset the data so that there are 5 evenly-spaced time points. In summary, the training data have 214 observations and the dimension for each observation is $5 \times 7$. The observation time is scaled to $[0, 1]$. Here, we fit the data by covariate GP-I-CFM, using the starting point as covariates, and generate 1000 LFP time series for each region (the starting LFP is generated from an I-CFM). For each second, the algorithm can run around 100 iterations per second (it runs around 2.5 iterations per second and takes longer time to converge when using all 50 time points). The results are shown in Figure 10. The generated time series can further be used to study neural activity in different brain regions. For example, the mean trajectories in Figure 10A suggest that the LFPs in Cornu Ammonis region 3 (CA3) and dentate gyrus (DG) are highly

correlated, which is consistent with the experiment fact that the rat DG does not project to any brain region other than the CA3 field of the hippocampus (Amaral et al., 2007). Besides this, we can use the generated samples to make more scientific and insightful conclusions. But this is beyond the scope of this paper.

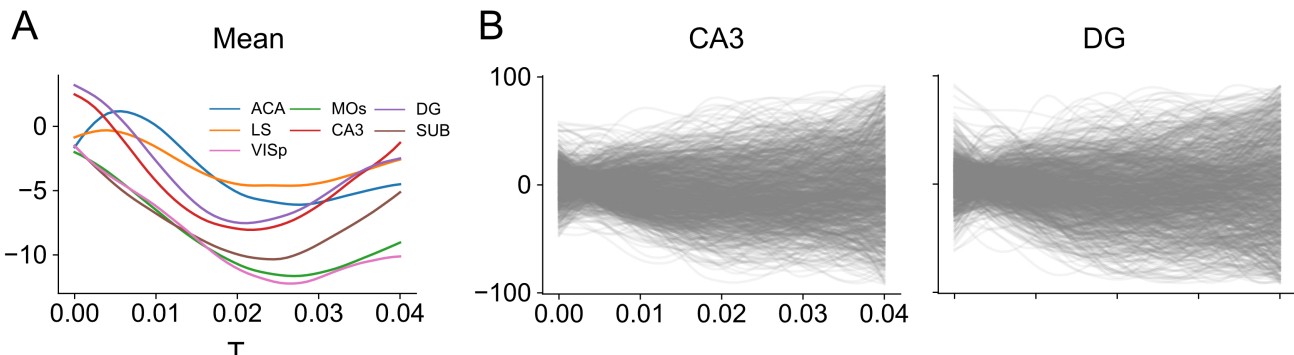

*Figure 10.* **Application to LFP data**.We apply the GP-I-CFM with covariate (on starting point) to a session of local field potential (LFP) data from 7 regions of mouse brain. In the training dataset, there are 214 observations (repeated trials). For each observation, it is a time series data of 5 time points from 7 brain regions. Here, we generate 1000 LFP time series for each region, where the starting LFP is generated from an I-CFM. **A.** The mean trajectories over 1000 samples. **B.** The generated 1000 time series for CA3 and DG.

## J. Proof of propositions

In this section, we provide proofs for several propositions in the main text. All these proofs are adapted from (Lipman et al., 2023; Tong et al., 2024).

### J.1. Proof for conditional FM on stream

**Proposition J.1.** *The marginal vector field over stream $u_t(x)$ generates the marginal probability path $p_t(x)$ from initial condition $p_0(x)$.*

*Proof.* Denote probability over stream as $q(\boldsymbol{s}) = \int p_{\boldsymbol{s}}(\boldsymbol{s} \mid x_0, x_1)\pi(x_0, x_1)d(x_0, x_1)$ and $p_t(x \mid \boldsymbol{s}) = \delta(x - s_t)$, then

$$\frac{d}{dt}p_t(x) = \frac{d}{dt}\int p_t(x \mid \boldsymbol{s})q(\boldsymbol{s})d\boldsymbol{s}$$

Assume the regularity condition holds, such that we can exchange limit and integral (and differentiation and integral) by dominated convergence theorem (DCT). Therefore,

$$= \int \frac{d}{dt}p_t(x \mid \boldsymbol{s})q(\boldsymbol{s})d\boldsymbol{s}$$

To handle the derivative on zero measure, define $s_t$-centered Gaussian conditional path and corresponding flow map as

$$p_{\sigma,t}(x \mid \boldsymbol{s}) := \mathcal{N}(x \mid s_t, \sigma^2 I)$$
$$\psi_{\sigma,t}(z \mid \boldsymbol{s}) := \sigma z + s_t,$$

for $z \sim N(0, I)$, such that $\lim_{\sigma \to 0} p_{\sigma,t}(x \mid \boldsymbol{s}) = p_t(x \mid \boldsymbol{s})$. Then by Theorem 3 of Lipman et al. (2023), the unique vector field defining $\psi_{\sigma,t}(z \mid \boldsymbol{s})$ (and hence generating $p_{\sigma,t}(x \mid \boldsymbol{s})$) is $u_t^*(x|\boldsymbol{s}) = ds_t/dt = u_t(s_t \mid \boldsymbol{s})$, for all $(t, x)$. Note that $u_t^*(x \mid \boldsymbol{s})$ extends $u_t(x \mid \boldsymbol{s})$ by defining on all $x$, and they are equivalent when $s_t = x$. Since $u_t^*(\cdot \mid \boldsymbol{s})$ generates $p_{\sigma,t}(\cdot \mid \boldsymbol{s})$,

by continuity equation,

$$\frac{d}{dt}p_t(x) = \int \frac{d}{dt} \lim_{\sigma \to 0} p_{\sigma,t}(x \mid \boldsymbol{s})q(\boldsymbol{s})d\boldsymbol{s}$$

$$= \int -\lim_{\sigma \to 0} \mathrm{div}(u_t^*(x \mid \boldsymbol{s})p_{\sigma,t}(x \mid \boldsymbol{s}))q(\boldsymbol{s})d\boldsymbol{s}$$

Then by DCT,

$$= -\lim_{\sigma \to 0} \mathrm{div}\left(\int u_t^*(x \mid \boldsymbol{s})p_{\sigma,t}(x \mid \boldsymbol{s})q(\boldsymbol{s})d\boldsymbol{s}\right)$$

$$= -\mathrm{div}\left(\int u_t^*(x \mid \boldsymbol{s}) \lim_{\sigma \to 0} p_{\sigma,t}(x \mid \boldsymbol{s})q(\boldsymbol{s})d\boldsymbol{s}\right)$$

$$= -\mathrm{div}\left(\mathbb{E}\left(u_t(x \mid \boldsymbol{s}) \mid s_t = x\right)p_t(x)\right)$$

By definition in equation 2,

$$= -\mathrm{div}\left(u_t(x)p_t(x)\right),$$

which shows that $p_t(\cdot)$ and $u_t(\cdot)$ satisfy the continuity equation, and hence $u_t(x)$ generates $p_t(x)$. $\qquad\square$

## J.2. Proof for gradient equivalence on stream

Recall

$$\mathcal{L}_{\mathrm{FM}}(\theta) = \mathbb{E}_{t,x}\|v_t^\theta(x) - u_t(x)\|^2,$$

$$\mathcal{L}_{\mathrm{sCFM}}(\theta) = \mathbb{E}_{t,\boldsymbol{s}}\|v_t^\theta(s_t) - u_t(x \mid \boldsymbol{s})\|^2,$$

where $x \sim p_t(x)$, $\boldsymbol{s} \sim q(\boldsymbol{s})$ and $q(\boldsymbol{s}) = \int p_{\boldsymbol{s}}(\boldsymbol{s} \mid x_0, x_1)\pi(x_0, x_1)d(x_0, x_1)$.

**Proposition J.2.** $\nabla_\theta \mathcal{L}_{FM}(\theta) = \nabla_\theta \mathcal{L}_{sCFM}(\theta)$.

*Proof.* To ensure existence of all integrals and to allow the changes of integral (Fubini's Theorem), we assume that $q(\boldsymbol{s})$ are decreasing to zero at a sufficient speed as $\|\boldsymbol{s}\| \to \infty$ and that $u_t$, $v_t$, $\nabla_\theta v_t$ are bounded. To facilitate proof writing, let $p_t(x \mid \boldsymbol{s}) = \delta(x - s_t)$.

The L-2 error in the expectation ca be re-written as

$$\|v_t^\theta(x) - u_t(x)\|^2 = \|v_t^\theta(x)\|^2 + \|u_t(x)\|^2 - 2\langle v_t^\theta(x), u_t(x)\rangle$$

$$\|v_t^\theta(s_t) - u_t(x \mid \boldsymbol{s})\|^2 = \|v_t^\theta(s_t)\|^2 + \|u_t(x \mid \boldsymbol{s})\|^2 - 2\langle v_t^\theta(s_t), u_t(x \mid \boldsymbol{s})\rangle$$

Thus, it's sufficient to prove the result by showing the expectations of terms including $\theta$ are equivalent.

First,

$$\mathbb{E}_x\|v_t^\theta(x)\|^2 = \int \|v_t^\theta(x)\|^2 p_t(x)dx$$

$$= \int \int \|v_t^\theta(x)\|^2 p_t(x \mid \boldsymbol{s})q(\boldsymbol{s})dxd\boldsymbol{s}$$

$$= \mathbb{E}_{\boldsymbol{s}} \int \|v_t^\theta(x)\|^2 \delta(x - s_t)dx$$

$$= \mathbb{E}_{\boldsymbol{s}}\|v_t^\theta(s_t)\|^2$$

Second,

$$
\begin{aligned}
\mathbb{E}_x \langle v_t^\theta(x), u_t(x) \rangle &= \int \langle v_t^\theta(x), u_t(x) \rangle p_t(x) dx \\
&= \int \langle v_t^\theta(x), \frac{\int u_t(x \mid \boldsymbol{s}) p_t(x \mid \boldsymbol{s}) q(\boldsymbol{s}) d\boldsymbol{s}}{p_t(x)} \rangle p_t(x) dx \\
&= \int \langle v_t^\theta(x), \int u_t(x \mid \boldsymbol{s}) p_t(x \mid \boldsymbol{s}) q(\boldsymbol{s}) d\boldsymbol{s} \rangle dx \\
&= \int \int \langle v_t^\theta(x), u_t(x \mid \boldsymbol{s}) \rangle \delta(x - s_t) q(\boldsymbol{s}) d\boldsymbol{s} dx \\
&= \mathbb{E}_{\boldsymbol{s}} \langle v_t^\theta(s_t), u_t(x \mid \boldsymbol{s}) \rangle
\end{aligned}
$$

These two holds for all $t$, and hence $\nabla_\theta \mathcal{L}_{\text{FM}}(\theta) = \nabla_\theta \mathcal{L}_{\text{sCFM}}(\theta)$ ☐

### J.3. Proof for gradient equivalence conditioning on covariates

Let $x$ be response, $c$ be covariates, and $\boldsymbol{s}$ be the stream connecting two endpoints $(x_0, x_1)$. Given covariate $c$, denote the conditional distribution of $\boldsymbol{s}$ as $q(\boldsymbol{s} \mid c) = \int p_{\boldsymbol{s}}(\boldsymbol{s} \mid x_0, x_1, c) \pi(x_0, x_1) d(x_0, x_1)$ and marginal conditional probability path as $p_t(x \mid c)$. Further, let

$$
\begin{aligned}
\mathcal{L}_{\text{cFM}}(\theta) &= \mathbb{E}_{t,x} \| v_t^\theta(x, c) - u_t(x \mid c) \|^2, \\
\mathcal{L}_{\text{cCFM}}(\theta) &= \mathbb{E}_{t,\boldsymbol{s}} \| v_t^\theta(s_t, c) - u_t(x \mid \boldsymbol{s}) \|^2,
\end{aligned}
$$

where $x \sim p_t(x \mid c)$ and $\boldsymbol{s} \sim q(\boldsymbol{s} \mid c)$

**Proposition J.3.** $\nabla_\theta \mathcal{L}_{\text{cFM}}(\theta) = \nabla_\theta \mathcal{L}_{\text{cCFM}}(\theta)$.

*Proof.* To ensure existence of all integrals and to allow the changes of integral (Fubini's Theorem), we assume that $q(\cdot \mid c)$ decreases to zero at a sufficient speed as $\|\boldsymbol{s}\| \to \infty$ and that $v_t^\theta$, $u_t$, $\nabla_\theta v_t^\theta$ are bounded. To facilitate proof writing, let $p_t(x \mid \boldsymbol{s}) = \delta(x - s_t)$.

The L-2 error in the expectations can be re-written as

$$
\begin{aligned}
\| v_t^\theta(x, c) - u_t(x \mid c) \|^2 &= \| v_t^\theta(x, c) \|^2 + \| u_t(x \mid c) \|^2 - 2 \langle v_t^\theta(x, c), u_t(x \mid c) \rangle \\
\| v_t^\theta(s_t, c) - u_t(x \mid \boldsymbol{s}) \|^2 &= \| v_t^\theta(s_t, c) \|^2 + \| u_t(x \mid \boldsymbol{s}) \|^2 - 2 \langle v_t^\theta(s_t, c), u_t(x \mid \boldsymbol{s}) \rangle
\end{aligned}
$$

Thus, it's sufficient to prove the result by showing the expectations of terms including $\theta$ are equivalent.

First,

$$
\begin{aligned}
\mathbb{E}_x \| v_t^\theta(x, c) \|^2 &= \int \| v_t^\theta(x, c) \|^2 p_t(x \mid c) dx \\
&= \int \int \| v_t^\theta(x, c) \|^2 p_t(x \mid \boldsymbol{s}) q(\boldsymbol{s} \mid c) dx d\boldsymbol{s} \\
&= \mathbb{E}_{\boldsymbol{s}} \int \| v_t^\theta(x, c) \|^2 \delta(x - s_t) dx \\
&= \mathbb{E}_{\boldsymbol{s}} \| v_t^\theta(s_t, c) \|^2
\end{aligned}
$$

Second,

$$
\begin{aligned}
\mathbb{E}_x \langle v_t^\theta(x, c), u_t(x \mid c) \rangle &= \int \langle v_t^\theta(x, c), u_t(x \mid c) \rangle p_t(x \mid c) dx \\
&= \int \langle v_t^\theta(x, c), \frac{\int u_t(x \mid \boldsymbol{s}) p_t(x \mid \boldsymbol{s}) q(\boldsymbol{s} \mid c) d\boldsymbol{s}}{p_t(x \mid c)} \rangle p_t(x \mid c) dx \\
&= \int \langle v_t^\theta(x, c), \int u_t(x \mid \boldsymbol{s}) p_t(x \mid \boldsymbol{s}) q(\boldsymbol{s} \mid c) d\boldsymbol{s} \rangle dx \\
&= \int \int \langle v_t^\theta(x, c), u_t(x \mid \boldsymbol{s}) \rangle \delta(x - s_t) q(\boldsymbol{s} \mid c) d\boldsymbol{s} dx \\
&= \mathbb{E}_{\boldsymbol{s}} \langle v_t^\theta(s_t, c), u_t(x \mid \boldsymbol{s}) \rangle
\end{aligned}
$$

These two holds for all $t$, and hence $\nabla_\theta \mathcal{L}_{\text{cFM}}(\theta) = \nabla_\theta \mathcal{L}_{\text{cCFM}}(\theta)$. $\square$

