# OpenReview forum: "Stream-level Flow Matching with Gaussian Processes"
_ICML.cc/2025/Conference — ICML 2025 poster_

### Official Review · Reviewer_R6jf · 2025-03-10

**Overall Recommendation:** 4

**Summary:**

This paper extends Conditional Flow Matching by introducing Gaussian processes to model latent "streams" connecting source and target distributions. Key contributions include: (1) a generalized CFM framework using GP streams while maintaining simulation-free training; (2) demonstrating reduced variance in vector field estimation and improved sample quality; (3) enabling flexible incorporation of correlated observations; and (4) empirical validation on synthetic, image, and time series datasets.

## update after rebuttal
The author resolved my confusion, so my current high rating remains unchanged.

**Claims And Evidence:**

The main claims are generally well-supported by both theoretical analysis and empirical evidence. The authors show (1) GP streams produce smoother vector fields with reduced variance; (2) GP-CFM improves sample quality across datasets;(3) The approach effectively incorporates correlated observations.

**Essential References Not Discussed:**

The paper covers most relevant literature.

**Experimental Designs Or Analyses:**

The experimental design is generally sound:

1. Synthetic examples effectively demonstrate the conceptual advantages.

2. Image generation experiments follow standard practices.

3. Time series experiments illustrate the unique capabilities of GP-CFM.

However, some experimental choices could be better justified:

1. The choice of hyperparameters for the GP kernels isn't thoroughly explained.

2. The comparison on CIFAR-10 could include more recent generative models beyond just CFM variants.

**Methods And Evaluation Criteria:**

The methods are sound and well-motivated. The authors evaluate their approach using standard metrics (FID, KID, Wasserstein distance) on appropriate datasets. Their comparison against baseline methods (I-CFM, OT-CFM) is reasonable, though they could potentially compare against more recent state-of-the-art generative models for broader context.

**Other Comments Or Suggestions:**

1. It would be better to include a more detailed comparison of computational requirements in the main text.

2. In the section of time series modeling, incorporating a discussion of Trajectory Flow Matching (Zhang et al., NeurIPS 2024) would provide useful context and highlight how your GP-based approach extends or differs from their method.

**Other Strengths And Weaknesses:**

**Strengths**:
1. The paper presents an theoretical framework for extending Conditional Flow Matching with Gaussian process streams.

2. The authors provide a novel perspective on variance reduction in flow matching through appropriate GP regularization.

3. The method shows valuable practical applications for time series and correlated data that standard CFM approaches cannot handle effectively.

4. The empirical results demonstrate clear improvements across multiple datasets as measured by standard quality metrics.

**Weaknesses**:
1. Computational complexity analysis could be more thorough.

2. Some hyperparameter choices need better justification,such as what principles should guide kernel selection and parameter tuning for different data modalities (images v.s. time series).

3. Potential limitations of the GP approach in very high dimensions, e.g, Imagenet64/256, aren't fully addressed.

**Questions For Authors:**

1. How does the computational complexity of GP-CFM scale with data dimensionality compared to standard CFM approaches?

2. What guidelines would you suggest for choosing appropriate GP kernels and their hyperparameters for different types of data?

**Relation To Broader Scientific Literature:**

The paper builds appropriately on previous work in flow matching (Lipman et al., 2023; Tong et al., 2024) and stochastic interpolants (Albergo et al., 2023, 2024). It extends CFM by incorporating Gaussian processes. The authors also make connections to Bayesian perspectives on generative modeling and optimally regularized estimation.

**Theoretical Claims:**

The theoretical foundations appear solid. I verified (1) the proof that the marginal vector field generates the probability path (Section J.1), (2) The gradient equivalence proofs (Sections J.2 and J.3), (3) the Bayesian decision-theoretic perspective (Appendix A).

---

> ### Author Rebuttal · Authors · 2025-03-30
>
> Thanks a lot to the reviewer for their positive comments, and thanks for suggestions on writing. We will update our manuscript accordingly, and add more details in the experiments (e.g. compare to more methods besides CFM variants) whenever possible. Here, we clarified some specific points…
>
> 1. > Potential limitations of the GP approach in very high dimensions, e.g, Imagenet64/256, aren't fully addressed.
>
> For computational convenience, we use independent GP for each dimension. Therefore, the proposed method should be scalable to high dimensional data. Even for many correlated data (e.g. long time seires), since the GP covariance is independent on observed values, we can calculate the covariance matrix inversion once before training. The current running time is reported with re-calculating matrix inversion for each training iteration. We will update the running time later.
>
>
> 2. > In the section of time series modeling, incorporating a discussion of Trajectory Flow Matching (Zhang et al., NeurIPS 2024) would provide useful context and highlight how your GP-based approach extends or differs from their method.
>
> Thanks for mentioning this paper. The time series application is one extension of our GP-CFM framework. The trajectory flow matching (Zhang et al., NeurIPS 2024) is based on autoregressive model (using FM to fit the AR functions), while we model the time seires by GP conditional stream. We will add this reference and the discussion in our camera-ready version if the paper is accepted.
>
> 3. > How does the computational complexity of GP-CFM scale with data dimensionality compared to standard CFM approaches?
>
> Since we use independent GP for each dimension, the computational bottleneck comes from the GP covariance matrix inversion over (artificial) time. However, the GP covariance and its inversion is independent on observed values, and hence we can calculate once before training. The current running time is reported with re-calculating matrix inversion for each training iteration. We will update the running time in later.
>
> 4. > What guidelines would you suggest for choosing appropriate GP kernels and their hyperparameters for different types of data?
>
> The choice of GP kernels may depend on prior information and constraint of problems. For example, if there’s a periodic pattern, we should chose the periodic kernel.
>
> For hyperparameter selection, it's a trade-off between systematic variance (from extrapolation error of neural net) and Monte Carlo error. The GP-stream reduces the systematic variance, but will introduce more Monte Carlo error. When implementing GP-CFM, we currently manually choose the GP parameter so that the GP conditional path covers a slightly wider region than linear interpolation (by checking several paired samples from the target to the source).
>
> To reduce the Monte Carlo error while preserving the reduction in systematic variance, instead of sampling one GP path, we can sample multiple GP paths or resort to importance sampling over t. We will add a discussion on tuning parameters in our camera-ready version if the paper is accepted.

---

> > ### Comment · Reviewer_R6jf · 2025-04-02
> >
> > The author has addressed my concern and I'll keep the score.

---

> > > ### Author Response · Authors · 2025-04-09
> > >
> > > Thank you very much again for your thoughtful comments and suggestions.

---

### Official Review · Reviewer_QyKi · 2025-03-11

**Overall Recommendation:** 2

**Summary:**

The paper proposes a novel flow matching method that incorporates **stochastic** bridges instead of **deterministic** bridges, which are typically used in the flow matching framework. In the context of generative modeling, flow matching (FM) is used to train neural ODEs with an initial distribution so that the distribution of their solutions matches a target (or terminal) distribution, commonly a data distribution. Specifically, FM enables the ODEs to learn and mimic the path measure (in a weak sense) defined by a collection of deterministic bridges, each connecting two points—one from the initial distribution and the other from the terminal distribution. A deterministic bridge is often chosen as a linear interpolation between two points over time. However, it is also possible to use a nonlinear deterministic bridge as long as it is path-wise continuously (time-)differentiable.

Unlike the typical FM, this paper proposes using stochastic bridges instead of deterministic ones. Here, stochastic bridges mean that for any joint sample pair from the initial and terminal distributions, there can be multiple time-dependent functions—i.e., sample paths (or streams, as termed in the paper)—that connect the two points. In order to generate such stochastic bridges, the paper proposes using Gaussian measures, as their sample paths are path-wise continuously (time-)differentiable, and the time derivative of the sample path has a closed-form solution. In particular, this property of Gaussian measures naturally facilitates their use within the flow matching framework.

It is important to note that the proposed method differs from Bridge matching, which relies on Brownian bridges and other Itô diffusion-based bridges while still employing a Markovian projection-style training approach similar to FM. In particular, sample paths generated by Itô diffusion-based bridges may not be time-differentiable in the conventional sense, even though they remain continuous. Nevertheless, the paper theoretically demonstrates that incorporating such stochastic bridges into flow matching is effective.

The authors demonstrate the efficacy of the proposed method on several benchmark datasets.

**Claims And Evidence:**

Overall, I find the paper novel and original. However, the motivation for using nonlinear stochastic bridges in the flow matching context is not entirely convincing, both theoretically and empirically.

For example, the use of a linear map (often referred to as the condOT path) is straightforward, and the discussion relating its linearity to reducing numerical errors provides a reasonable justification—particularly in how it accelerates the solving of ODEs with fewer evaluation steps. However, in comparison to this widely adopted approach, the motivation for introducing nonlinear bridges remains unconvincing despite its crucial role in the proposed method. Similarly, making the bridges stochastic does not seem to offer a clear advantage, and its justification is not well-supported.

In this regard, the experimental results also appear somewhat limited. While the authors provide several comparative results on popular image generation datasets such as MNIST and CIFAR-10, the performance differences between the proposed method and existing approaches do not appear substantial. The reported improvements, if any, seem marginal, making it difficult to assess the practical advantages of using nonlinear and stochastic bridges. Furthermore, it remains unclear whether the observed gains, if they exist, are due to the proposed modifications or other confounding factors. It would be helpful if the authors could provide stronger empirical evidence or further theoretical insights to clarify the benefits of using nonlinear/stochastic bridges of this framework.

**Essential References Not Discussed:**

N/A

**Experimental Designs Or Analyses:**

See “Claims And Evidence”

**Methods And Evaluation Criteria:**

N/A

**Other Comments Or Suggestions:**

Once again, I find the paper novel and original. As mentioned earlier, it is also distinctive even when compared to Bridge matching. In addition, I appreciate how this work broadens my perspective on flow matching, and I found it an enjoyable read.

However, aside from its novelty and originality, the experimental results did not fully convince me. I believe the chosen tasks may not be well-suited to the proposed method. There could be more relevant applications where the sample paths of the generation process need to be controlled in specific ways, which might better showcase the advantages of the approach and strengthen its motivation.

Additionally, while it may not be strictly necessary, it could be helpful to explicitly clarify how this method differs from other Bridge matching approaches. At first glance, I initially found the two somewhat confusing due to their use of stochastic interpolation paths (or streams), and I believe other readers might have a similar impression. Addressing this distinction more clearly could enhance the paper’s accessibility and impact.

**Other Strengths And Weaknesses:**

See "Other Comments Or Suggestions"

**Questions For Authors:**

N/A

**Relation To Broader Scientific Literature:**

N/A

**Theoretical Claims:**

See “Claims And Evidence”

---

> ### Author Rebuttal · Authors · 2025-03-30
>
> Thanks a lot to the reviewer for their positive comments, and thanks for suggestions on writing. We will update our manuscript accordingly, and add more details in the experiments whenever possible. Here, we clarified some specific points…
>
> 1. > However, in comparison to this widely adopted approach, the motivation for introducing nonlinear bridges remains unconvincing despite its crucial role in the proposed method. Similarly, making the bridges stochastic does not seem to offer a clear advantage, and its justification is not well-supported.
>
> The stochastic interpolant was previously studied theoretically in [1], where the benefits of stochastic interpolant was discussed in section 4.3 of their paper. In their paper, they demonstrated that the stochastic interpolant suppresses spurious intermediate, which smooth path and vector field (Figure 7 in our paper). Therefore, stochastic interpolant can simplify estimation and accelerate the ODE integration.
>
> In our paper, we further extend the “stochastic interpolant” in [1], since the Brownian bridge used in [1] is a special case of a Gaussian process (GP), and therefore, conditioning on the entire stream and modeling it using a GP offers greater modeling flexibility. Furthermore, by resorting to properties of GP (conditional distribution and derivative of a GP are still GPs), we preserve the simulation-free property of FM algorithm. Besides, we further provide a bias-variance trade-off perspective for advantages of stochastic interpolation.
>
>
> 2. > It could be helpful to explicitly clarify how this method differs from other Bridge matching approaches.
>
> The connection to bridge (e.g.  Schr\"odinger bridge) and other related models are discussed in [1]. Specifically, in section 3.4, they showed that the stochastic interpolants can recover the Schr\"odinger bridge (from source to target densities), if we explicitly optimize over the interpolant. We will add more discussions on this in the updated manuscript.
>
> [1] Stochastic Interpolants: A Unifying Framework for Flows and Diffusions, Michael S. Albergo, Nicholas M. Boffi, Eric Vanden-Eijnden, 2023

---

> > ### Comment · Reviewer_QyKi · 2025-04-04
> >
> > The authors have partially addressed my concerns, but I remain somewhat unconvinced on a few points, so I would prefer to keep my original score.
> >
> > For example, the authors provide a one-dimensional experiment to support the claim that stochastic interpolants are beneficial. Unfortunately, I do not believe this level of experimentation sufficiently supports the general statement that *“all stochastic interpolants help smooth the generation path.”* If the authors aim to make such a broad claim, experiments on large-scale datasets, where this property would be of real interest, would have been more appropriate.
> >
> > In addition, the authors cite the phrase *“stochastic interpolant suppresses spurious intermediate ...”* from a relevant work to support the claim as well. If the authors believe that their proposed stochastic interpolant is effectively the same as prior stochastic interpolants—and that the smoother integration path is therefore a general property—this would actually diminish the novelty of the paper. Conversely, if the analysis is truly specific to the differentiable Gaussian measure used in this work, then the conclusions should not be presented as broadly applicable to stochastic interpolants in general.

---

> > > ### Author Response · Authors · 2025-04-09
> > >
> > > Thank you for your thought-provoking comments and suggestions. We acknowledge that the one-dimensional experiment is indeed limited and we would love to carry out more extensive, multi-dimensional numerical experiments in the future to substantiate our discussion. Regarding the novelty of our proposed method, while our GP strategy inherits some general properties of stochastic interpolates, it also enjoys some unique properties that come from properties of GPs that general stochastic interpolants do not enjoy. We believe the novelty in our approach lies in exploiting these unique properties of GPs to construct a computational efficient and robust extension to the CFM algorithm. We hope we will be able to make our argument more convincing in future revisions. We very much appreciate your time and consideration. Thank you again.

---

### Official Review · Reviewer_23pK · 2025-03-17

**Overall Recommendation:** 3

**Summary:**

This paper proposes a generalization of conditional flow matching (CFM) models using Gaussian process (GP) streams. While CFM uses two endpoints as condition, GP stream defines a GP over time that connects 2 or more points from $t=0$ to $t=1$, providing more controls over the mean and variance of the path (thus providing stronger regularization) and enabling time-series modeling.

**Claims And Evidence:**

Yes. The claimed contributions appear are supported by the theoretical analysis and experiments.

**Essential References Not Discussed:**

Essential references have been discussed.

**Experimental Designs Or Analyses:**

Some aspects of the experimental setup are unclear. In Section 4.1, how is the GP specifically constructed? Please provide an equation or reference the appendix. Is a GP stream connecting two endpoints the same as or different from stochastic interpolant ($\sigma>0$)? Why would noise-free GP streams still provide stronger regularization (my understanding is that only the mean is altered)?

In Fig. 7, the comparison does not seem fair because I-CFM and GP-CFM appear noise-free, whereas their GP counterparts appear stochastic.

**Methods And Evaluation Criteria:**

The method is derived from a theoretical perspective and is a reasonable extension to flow matching. The scale of the experiments is small but reasonable for a theory-focused paper.

**Other Comments Or Suggestions:**

The y-axis label in Fig. 2 is missing, which should be $x$ I think.

**Other Strengths And Weaknesses:**

The paper is challenging to read as the writing is also unclear in some parts. For example, in Section 3.2, there is no formal definition of the Gaussian process in equation form in the main text. Although it is presented in Appendix C, it would be better to include a simplified equation in the main text (e.g., assuming a diagonal covariance) or at least provide a direct reference to the appendix. Additionally, several notations are not explained, such as $m(t)$ and $I_d$.

**Questions For Authors:**

Although the GP stream enables time-series modeling, an alternative approach is to treat it not as a time-series flow but as a joint distribution over multiple frames (e.g., as in video diffusion models). Is it possible to model videos as a time-series flow? Would this offer any actual advantages over the current paradigm? It seems to me that the additional conditioning (covariate) would introduce extra complexity, making it more similar to auto-regressive modeling. Speaking of auto-regressive modeling, would time-series modeling using a GP suffer from drifting (error accumulation)?

**Relation To Broader Scientific Literature:**

The proposed method is a theoretical extension to CFM, with finer control over the flow paths. The finding that adding noise to the path improves sample quality due to stronger regularization and less over-fitting is already observed in prior work.

**Theoretical Claims:**

The theoretical claims appear correct. The proofs in the Appendix are adapted from established prior work.

---

> ### Author Rebuttal · Authors · 2025-03-30
>
> Thanks a lot to the reviewer for their positive comments, and thanks for suggestions on writing. We will update our manuscript accordingly. Here, we clarified some specific points…
>
> 1. > Some aspects of the experimental setup are unclear.
>
> The details of GP construction can be found in Appendix C and E.
>
> The GP interpolant is different from stochastic interpolant referred here (linear interpolant with $\sigma > 0$). To visualize the difference, please refer to Figure 2 and 8. The $\sigma$ adds random jitters to the interpolation, but GP stream allows the interpolant oscillate smoothly. We can design $\sigma(t)$ to further help the GP interpolant, as shown in the Figures.
>
> In figure 7, we set $\sigma=10^{-3}$ for all four algorithms.
>
>
> 2. > Although the GP stream enables time-series modeling, an alternative approach is to treat it not as a time-series flow but as a joint distribution over multiple frames (e.g., as in video diffusion models). Is it possible to model videos as a time-series flow? Would this offer any actual advantages over the current paradigm?
>
> Yes, we can model the time-series via joint distribution over multiple frames, but we may need to model in latent space and factorize the spatial and temporal components of model architecture. Otherwise, the dimension is too high to be feasible for applications such as video generation. Using the GP-CFM for time series modeling explicitly models the correlation over multiple correlated samples into one unifying model, without increasing the dimension linearly with number of time points, in the context of time series.
>
> 3. > It seems to me that the additional conditioning (covariate) would introduce extra complexity, making it more similar to auto-regressive modeling. Speaking of auto-regressive modeling, would time-series modeling using a GP suffer from drifting (error accumulation)?
>
> In our paper, adding additional covariates (labels) is related to AR modeling, but not exactly the same. Here, we use the starting point at $t = 0$ as covariates, which is static along the time. Therefore, there's no drifting issue. It's possible to use past lag-p observations  as covariates to be AR model, but this would make problems more complicated, and as mentioned by the reviewer, we need to concern about drifting issue.

---

> > ### Comment · Reviewer_23pK · 2025-04-09
> >
> > Thank you for providing additional clarifications. If my understanding is correct, the primary distinction between GP-CFM and the baseline methods lies in their covariance kernels: the baselines employ white noise kernels, resulting in time series composed of independent samples, whereas GP-CFM generalizes to kernels beyond white noise, thus capturing temporal dependencies.
> >
> > However, I'm still not convinced of how this generalization inherently leads to stronger regularization. It seems that by simply adjusting the time-varying variance within the stochastic interpolant baseline, one could achieve marginal distributions similar to GP-CFM at any time slice. For instance, taking Figure 7 as an example, by increasing the variance ($\sigma$) around $t=0.5$, the histogram produced by I-CFM could resemble that of GP-I-CFM. Thus, I feel that the comparison is still not entirely fair.

---

> > > ### Author Response · Authors · 2025-04-09
> > >
> > > If we simply replace the kernel in the GP-CFM with a white noise (which is referred to as nuggets in the GP literature). What we would get would be a non-smooth oscillating stochastic path that connects the end points in the CFM algorithm. Such a path is generally not differentiable and hence would not enjoy the close-form, GP derivative property and therefore will not lead to a simple algorithm such as our GP-CRM. Now if in addition, we reduce the nugget variance down to zero, which removes the stochasticity entirely from the GP-CFM, then yes the GP-CFM will reduce down to the baseline model such as I-CFM or the OT-I-CFM depending on how the endpoints are coupled.
> > >
> > > An important property of GP-based time-series in comparison to many AR-based models is that GP-based times-series does not require the observations to be lying on shared, equal-spaced time points. In fact, observations can even lie on irregular intervals that are unique to each path, and the GP provides an effective modeling of the correlation structure over the shared observations. This property is important in the context of CFM training, in that we don’t want to restrict ourselves to applications that always share the time slices. Some time slices may have more or less points than others and some subjects may have more or less observations over time than others.
> > >
> > > Additionally, while the GP approach does not accumulate errors over time as AR-based models since there is not a natural ordering of time in GP-modeling, there is indeed a limitation in the GP approach compared to AR-based time-series modeling, however, which may limit its effectiveness in video modeling as time-series. It is that GP-based time-series modeling generally attempts to use all observations in the entire time domain to model the transition of observations overtime, (in contrast, AR-based models using only one or a small number of previous time points to model the next,) and thus can be ineffective in capturing drastic changes in adjacent time frames. This can create visually blurry transitions in some frames.
> > >
> > > Because videos are represented as frames of images over evenly spaced time grid, AR-based models can in fact be very effective. At the same time, the referee has pointed to an excellent direction to explore GP modeling in this regard as well. We believe it is possible to create a hybrid of AR and GP modeling that enjoys the unique benefits of each. This may lead to a very effective model for videos. We would love to explore this direction in the future.
> > >
> > > Many thanks again for your thoughtful comments and excellent suggestions!

---

### Official Review · Reviewer_3C93 · 2025-03-18

**Overall Recommendation:** 4

**Summary:**

The paper introduces stream-level flow matching with Gaussian processes (GP-CFM), which extends conditional flow matching to matching streams, i.e. latent stochastic paths that connect the source and target end points using Gaussian processes. The proposed framework naturally allows to include correlated observations (e.g. time series date) while remaining simulation-free, as the position and velocity can be readily sampled from the GP.

**Claims And Evidence:**

The claims and evidence, experiments, and arguments provided to validate the claims are generally sound (more details below). The authors demonstrate the workings of their method on synthetic data, which adds to the overall exposition, and evaluate its utility on multiple standard datasets. While none of the considered datasets are inherently high dimensional or used to compare state-of-the-art image generation with flows, it sufficiently demonstrates the utility of the proposed framework in multiple settings.

**Variance reduction in the estimation of the marginal vector field leads to improved sample quality.**

The authors show that using the GP-stream variants of CFM results in improved sample quality in terms of lower average Wasserstein-2 distance (synthetic example), FID (MNIST, CIFAR10), and KID (MNIST). However, the authors do not directly access estimator variance and I don't see how the improvement in average Wasserstein-2 distance (or FID) can be directly attributed to a lower (per stream) estimator variance. Without further assumptions it could also be attributed to lower bias. If the assumption is that the estimator of the marginal vector field is unbiased, I can see how an improvement must be the result of a reduction in estimator variance. However, in practice we don't have access to the true posterior probability path required to estimate the marginal vector field (Equation 2 in the manuscript), but have to resort to the learned approximation, which leads to a biased estimator.

Can the authors please clarify this point (see questions)?

**GP stream variants can naturally accommodate correlated observations.**

The authors first demonstrate how GP-steam variants can leverage multiple training observations using 2d synthetic examples. They further show the utility of their framework in the time series setting on data from the LFP dataset and synthetic data, corresponding to modifying digits from HWD+. The latter shows significantly better FID scores, w.r.t the correct digit distributions, across time.

**Essential References Not Discussed:**

All essential references are included.

**Experimental Designs Or Analyses:**

I have read the experiment section and have validated the soundness of the experiments.

**Methods And Evaluation Criteria:**

Yes.

**Other Comments Or Suggestions:**

None.

**Other Strengths And Weaknesses:**

### Strengths

- The paper presents a novel extension to Conditional Flow Matching, which is mathematically elegant and offers several advantages:
- The approach provides a principled way to model the uncertainty in flow paths.
- The GP formulation remains "simulation-free"
- The ability to incorporate multiple correlated observations into a unified model
- The framework is complementary to existing methods like OT-CFM.

### Weaknesses

The computational overhead of the GP calculations and their implication should be discussed in more detail. While the authors mention "moderate computational cost," a more detailed discussion of the computational complexity, especially in the case of high-dimensional data (without independence assumption) and multiple correlated observations, would be valuable.

**Questions For Authors:**

1. Regarding the reduction in estimator variance
- Can you please clarify how the improvement in Wasserstein-2 distance (or FID, KID) relates to a reduction in estimator variance opposed to a possible reduction in bias?
- Did you derive the true posterior probability to report the results for the synthesis 2D 2-Gaussian mixture?
2. While broadening the coverage region reduces problems associated with extrapolation it intuitively also seems to increase the data demands to robustly learn a model covering the broader coverage region. Is this a potential concern for real application? Are there practical guidelines on how practitioners should choose/tune their covariance kernels to achieve a good amount of coverage while extending it too much

**Relation To Broader Scientific Literature:**

The authors extend prior work from Lipman et al. [1] and Tong et al. [2], which specify conditional probability paths by defining a reference vector field given one or both endpoints respectively, by instead specifying a stochastic process (specifically a GP) that connects these endpoints.

The authors appropriately cite relevant work on conditional flow matching and cite Rasmussen & Williams [3] for their fundamental work on GPs. While not essential I think the authors should make an effort to also point out existing work that aims to directly model ODEs using GPs, and to discuss the conceptual differences to these approaches.

[1] Lipman et al. Flow Matching for Generative Modeling. ICLR, 2023.
[2] Tong et al. Improving and generalizing flow-based generative models with mini-batch optimal transport. TMLR, 2024
[3] Rasmussen and Williams, Gaussian Processes for Machine Learning. MIT Press, 2005

**Theoretical Claims:**

I have not checked any proofs.

---

> ### Author Rebuttal · Authors · 2025-03-30
>
> Thanks a lot to the reviewer for their positive comments. Here, we clarified some specific points…
>
> 1. > However, the authors do not directly access estimator variance and I don't see how the improvement in average Wasserstein-2 distance (or FID) can be directly attributed to a lower (per stream) estimator variance. Can you please clarify how the improvement in Wasserstein-2 distance (or FID, KID) relates to a reduction in estimator variance opposed to a possible reduction in bias?
>
> We thank the reviewer for pointing this out, which has helped us clarify the sources of bias and variance in our algorithms.
>
> There are two sources of variance: 1) systematic variance from extrapolation of the neural network, 2) Monte Carlo variance of vector field estimation. The GP-stream reduces the systematic variance by expanding the search region, but the stochastic interpolation introduces more Monte Carlo error. To reduce the Monte Carlo error, instead of sampling one GP path, we can sample multiple GP paths at the same time.
>
> To validate the argument above, we did a quick experiment. We considered the target to be a 1D 2-Gaussian mixture, and drew 200 samples for training. We tried I-CFM, GP-I-CFM with 1 GP path (GP(1)-I-CFM) and 10 GP paths (GP(10)-I-CFM). We repeated the training 30 times and generated 10,000 samples for each. We smoothed the generated samples by Gaussian kernel density estimation (KDE). The means of the KDE match the true density well for all three algorithms. However, compared to I-CFM, the standard deviation of the KDE lines is lower in GP(1)-I-CFM (oscillates less around the true density), and GP(5)-I-CFM further reduces the standard deviation. To summarize results, we calculate mean of $|\overline{kde}(x) - f(x)|$ and $s_{kde}(x)$, where $x$ are 1000 evenly spaced points over a $(-6, 6)$. Denote $\overline{\lvert \overline{kde}(x_i) - f(x_i) \rvert} = \frac{1}{1000}\sum_{i=1}^{1000} \lvert \overline{kde}(x_i) - f(x_i) \rvert$ and $\overline{s_{kde}(x_i)} = \frac{1}{1000}\sum_{i=1}^{1000} s_{kde}(x_i)$
>
> |             | $\overline{\lvert \overline{kde}(x_i) - f(x_i) \rvert}$ | $\overline{s_{kde}(x_i)}$ |
> |-------------|-----|-----|
> | I-CFM       |   0.0035  |  0.0154   |
> | GP(1)-I-CFM    |   0.0036  |  0.0146   |
> | GP(5)-I-CFM|   0.0036  |  0.0136   |
>
> Since we are not allowed to update the manuscript or upload figures in the rebuttal, we will include more detailed experiments and discussions in the updated manuscript.
>
> 2. > Did you derive the true posterior probability to report the results for the synthesis 2D 2-Gaussian mixture?
>
> No, we didn't. For GP-I-CFM of 2D 2-Gaussian mixture, we can derive the marginal probability path. However, here we are considering the quality of generated samples, and checking the bias and variance of generated samples at $t=1$ plays the same role as checking the whole sample path from $t=0$ to $t=1$.
>
> 3. > While broadening the coverage region reduces problems associated with extrapolation it intuitively also seems to increase the data demands to robustly learn a model covering the broader coverage region. Is this a potential concern for real application? Are there practical guidelines on how practitioners should choose/tune their covariance kernels to achieve a good amount of coverage while extending it too much.
>
> If we sample 1 GP path for each iteration, it's a trade-off between systematic variance and Monte Carlo error (as mentioned above). When implementing GP-CFM, we currently manually choose the GP parameter so that the GP conditional path covers a slightly wider region than linear interpolation (by checking several paired samples from the target to the source).
>
> To reduce the Monte Carlo error while preserving the reduction in systematic variance, instead of sampling one GP path, we can sample multiple GP paths or resort to importance sampling over $t$. We will add a discussion on tuning parameters in our camera-ready version if the paper is accepted.

---

### Decision · Program_Chairs · 2025-05-01

**Decision:**

Accept (poster)

**Comment:**

This paper proposes using Gaussian processes to model the latent stochastic paths between draws from the coupling that arise in flow matching.
This enables finer control over the regularization of the path and thus provides variance reduction and is more suited for time-series and correlated data.
The reviewers found the work to be original and a valuable theoretical contribution. The proposed approach is readily compatible with other variance
reduction approaches that define the choice of coupling (e.g. minibatch OT) and opens up a new design space regarding the kernel design for the GP.
While the experimental validation of the proposed approach
is somewhat limited and the paper would benefit by benchmarking on higher-dimensional datasets, the positives outweigh the drawbacks.